# Language Models over Canonical Byte-Pair Encodings

**Tim Vieira** [1]  **Tianyu Liu** [1]  **Clemente Pasti** [1]  **Yahya Emara** [1]  **Brian DuSell** [1]  **Benjamin LeBrun** [2]
**Mario Giulianelli** [1]  **Juan Luis Gastaldi** [1]  **Timothy J. O'Donnell** [3 2 4]  **Ryan Cotterell** [1]

## Abstract

Modern language models represent probability distributions over character strings as distributions over (shorter) token strings derived via a deterministic tokenizer, such as byte-pair encoding. While this approach is highly effective at scaling up language models to large corpora, its current incarnations have a concerning property: the model assigns nonzero probability mass to an exponential number of *noncanonical* token encodings of each character string——these are token strings that decode to valid character strings but are impossible under the deterministic tokenizer (i.e., they will never be seen in any training corpus, no matter how large). This misallocation is both erroneous, as noncanonical strings never appear in training data, and wasteful, diverting probability mass away from plausible outputs. These are avoidable mistakes! In this work, we propose methods to enforce canonicality in token-level language models, ensuring that only canonical token strings are assigned positive probability. We present two approaches: (1) canonicality by conditioning, leveraging test-time inference strategies without additional training, and (2) canonicality by construction, a model parameterization that guarantees canonical outputs but requires training. We demonstrate that fixing canonicality mistakes improves the likelihood of held-out data for several models and corpora.

[Github] github.com/genlm/canonical-icml-2025

## 1. Introduction

Modern language models are probability distributions over character strings (denoted $\Sigma^*$) that are parameterized as

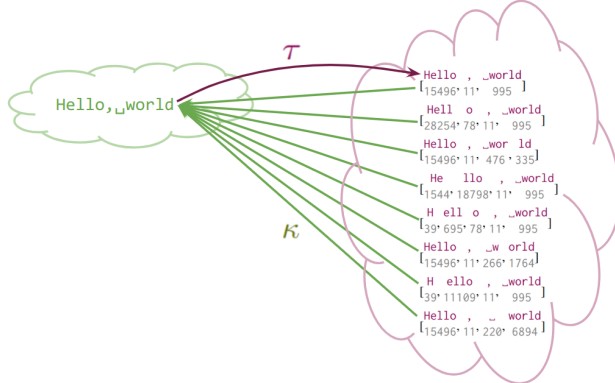

*Figure 1.* The figure shows the canonical and noncanonical encodings of the string `Hello,␣world`. The diagram shows the top-8 token encodings of the string according to their probability (descending top to bottom), as there are hundreds of them for this short string. Note that the canonical token encoding is the most likely one (i.e., it is at the top), which is reassuring as it is the most representative of the training data.

distributions over a different space of *token* strings (denoted $\Delta^*$). In the case of the prevalent byte-pair encoding (BPE; Gage, 1994), each token may be regarded as a multi-character (or multi-byte) chunk of a UTF-8-encoded string. Such a language model is trained by *encoding* its training data into token strings, which has the benefit of compressing the large corpus. The BPE encoding is a pair of functions: an encoding function $\tau\colon \Sigma^* \to \Delta^*$ that maps a given input character string $\boldsymbol{\sigma}$ to its *canonical encoding* $\boldsymbol{\delta} = \tau(\boldsymbol{\sigma})$, and a decoding function $\kappa\colon \Delta^* \to \Sigma^*$ that maps any token string $\boldsymbol{\delta}$ back to a character string $\boldsymbol{\sigma} = \kappa(\boldsymbol{\delta})$. The result of training on a corpus of token strings is a language model $p_\Delta$ that generates a token string $\boldsymbol{\delta}$ and decodes it to a character string $\boldsymbol{\sigma}$.

We define the set of *canonical* token strings $\mathcal{D} \subseteq \Delta^*$ as those that appear as the canonical encoding of at least one character string, i.e., $\mathcal{D} \stackrel{\text{def}}{=} \{\tau(\boldsymbol{\sigma}) \mid \boldsymbol{\sigma} \in \Sigma^*\}$. These are the only strings that are *possible* in the tokenized training data of $p_\Delta$. The set of *noncanonical* token strings is, thus, $\Delta^* \smallsetminus \mathcal{D}$. Even though $p_\Delta$ has never—and will never—see a noncanonical token string in its training data, it will still place nonzero probability mass on them due to model architecture and training limitations. Fig. 1 shows an example

---

[1]ETH Zürich  [2]Mila  [3]McGill University  [4]Canada CIFAR AI Chair. Correspondence to: Tim Vieira <tim.f.vieira@gmail.com>.

*Proceedings of the 42^nd International Conference on Machine Learning*, Vancouver, Canada. PMLR 267, 2025. Copyright 2025 by the author(s).

of a string with its canonical encoding and its exponentially many noncanonical encodings.

**Fixing canonicality mistakes.** Removing noncanonical strings from the support of the estimated distribution $p_\Delta$ can improve the likelihood of the correct token strings. We describe two families of methods for doing so

- Canonicality by conditioning: We explore efficient test-time inference methods for conditionally generating text that satisfies the canonicality constraint without retraining.
- Canonicality by construction: We explore methods that impose canonicality constraints directly in the language model's parameterized architecture and give a method to fine-tune its parameters.

In addition to these novel methods, this paper presents the following contributions:

- We prove that our methods can only improve the fit to the true distribution over tokens.
- We show empirically that fixing canonicality mistakes using our methods improves the likelihood of held-out data for several models and corpora.
- As a side effect of pursuing the goal of generating only canonical token strings, we discovered an efficient, incremental canonicality test for BPE (see App. B) that is significantly simpler than prior work (e.g., Berglund et al., 2024; Cognetta & Okazaki, 2024), as it does not require automata theory to understand or implement.

## 2. Preliminaries

### 2.1. Strings

An **alphabet** is a non-empty, finite set of **symbols** (e.g., bytes, characters, or tokens) from which we can build strings. A **string** is a finite-length sequence of symbols from a given alphabet. Let $\Gamma$ be an alphabet. Let $\Gamma^*$ denote the **set of all strings** over $\Gamma$. We use $|\gamma|$ to denote the length of the string and $\varepsilon$ to denote the string of zero length. Let $\gamma' \cdot \gamma''$ denote the concatenation of $\gamma'$ and $\gamma''$. We write $\gamma_{<n} \overset{\text{def}}{=} \gamma_1 \cdots \gamma_{n-1}$ to access the first $n-1$ symbols of the string $\gamma$. Let $\gamma \preceq \gamma'$ denote that $\gamma$ is a prefix of $\gamma'$, and $\gamma' \succeq \gamma$ denote that $\gamma'$ is has $\gamma$ as a prefix. The **set of bigrams** of a given string $\text{BIGRAMS}(\gamma_1 \cdots \gamma_N) \overset{\text{def}}{=} \{\gamma_n \gamma_{n+1} \mid 1 \le n < N\}$.

### 2.2. Information-Theoretic Quantities

For completeness, we include definitions of the following common information-theoretic quantities, which are measured in bits. Let $\log$ be the logarithm function in base 2 throughout this paper. Let $p$ and $q$ be probability distributions over a countable domain $\mathcal{X}$.

- **Entropy**: $\text{H}(p) \overset{\text{def}}{=} -\sum_{x \in \mathcal{X}} p(x) \log p(x)$

- **Cross-entropy**: $\text{H}(p, q) \overset{\text{def}}{=} -\sum_{x \in \mathcal{X}} p(x) \log q(x)$
- **KL divergence**: $\text{KL}(p\|q) = \sum_{x \in \mathcal{X}} p(x) \log \frac{p(x)}{q(x)}$

These quantities are related via $\text{KL}(p\|q) = \text{H}(p, q) - \text{H}(p)$, and $\text{H}(p) = \text{H}(p, p)$. Lastly, we follow the information-theoretic convention that $0 \log 0 \overset{\text{def}}{=} 0$ in these definitions.

### 2.3. Language Models

A language model $p_\Gamma$ is a probability distribution over a set of strings $\Gamma^*$. Thus, we can define a string-valued random variable $Y \sim p_\Gamma$ with this distribution. Moreover, we can compute the conditional probabilities of different events pertaining to the random variable $Y$, as we normally do in probability theory. However, since $Y$ is string-valued, the comparisons and transformations applied to it must be string-friendly. We define the **prefix probability**:

$$\overrightarrow{p_\Gamma}(\gamma) \overset{\text{def}}{=} \Pr_{Y \sim p_\Gamma}[Y \succeq \gamma] = \sum_{\gamma' \in \Gamma^*} \mathbb{1}\{\gamma' \succeq \gamma\} \, p_\Gamma(\gamma') \quad (1a)$$

and the **conditional prefix probability**:

$$\overrightarrow{p_\Gamma}(\gamma' \mid \gamma) \overset{\text{def}}{=} \Pr_{Y \sim p_\Gamma}[Y \succeq \gamma \cdot \gamma' \mid Y \succeq \gamma] = \frac{\overrightarrow{p_\Gamma}(\gamma \cdot \gamma')}{\overrightarrow{p_\Gamma}(\gamma)} \quad (1b)$$

$$\overrightarrow{p_\Gamma}(\text{EOS} \mid \gamma) \overset{\text{def}}{=} \frac{p_\Gamma(\gamma)}{\overrightarrow{p_\Gamma}(\gamma)} \quad (1c)$$

We can use conditional prefix probability to factorize the probability $p_\Gamma(\gamma)$ of a string $\gamma = \gamma_1 \cdots \gamma_T$ as the conditional probability of each of its symbols $\gamma_1, ..., \gamma_T$, followed by an **end-of-string event** EOS:[1]

$$p_\Gamma(\gamma) = \overrightarrow{p_\Gamma}(\text{EOS} \mid \gamma) \prod_{t=1}^{|\gamma|} \overrightarrow{p_\Gamma}(\gamma_t \mid \gamma_{<t}) \quad (1d)$$

Note that each conditional prefix distribution $\overrightarrow{p_\Gamma}(\cdot \mid \gamma_{<t})$ is a probability distribution over $\Gamma \cup \{\text{EOS}\}$. Many language models are *defined* via the product in Eq. (1d) where each single-symbol conditional prefix probability comes from a parametric model trained by predicting the next token given the previous context. Examples of this include transformers (Vaswani et al., 2017), RNNs (e.g., Mikolov et al., 2010; Sundermeyer et al., 2015), and $n$-gram models (e.g., Shannon, 1948).

### 2.4. Tokenized Language Models

An **(exact, deterministic) tokenization model** (Gastaldi et al., 2025) is a tuple $(\Sigma, \Delta, \tau, \kappa)$ where $\Sigma$ is an alphabet of characters (or bytes), $\Delta$ is an alphabet of tokens, $\tau \colon \Sigma^* \to \Delta^*$ is an **encoding function**, $\kappa \colon \Delta^* \to \Sigma^*$ is a **decoding function**, and $\kappa(\tau(\sigma)) = \sigma$ for all $\sigma \in \Sigma^*$.[2]

---

[1]Here, EOS is an event, *not* a symbol; thus, it may never appear in any well-formed string.

[2]Note that for $\kappa$ and $\tau$ to form a *bijection* between $\Sigma^*$ and $\Delta^*$, we would additionally require $\tau(\kappa(\delta)) = \delta$ for all $\delta \in \Delta^*$.

A **tokenized language model** is a language model $p_\Sigma$ over character strings $\Sigma^*$ that is *parameterized* by a token-level language model $p_\Delta$ over $\Delta^*$ and a decoding function $\kappa \colon \Delta^* \to \Sigma^*$. The tokenized language model defines the following probability distribution over $\Sigma^*$:

$$p_\Sigma(\boldsymbol{\sigma}) \stackrel{\text{def}}{=} \sum_{\boldsymbol{\delta} \in \Delta^*} \mathbb{1}\{\boldsymbol{\sigma} = \kappa(\boldsymbol{\delta})\}\, p_\Delta(\boldsymbol{\delta}) \tag{2}$$

Notice that $\tau$ is conspicuously absent from this expression, as it is only part of the estimation process, which we characterize below. In §4, we consider alternatives to Eq. (2) that exploit prior knowledge of $\tau$ to force the probability of noncanonical token strings to be zero.[3]

**Estimating a tokenized language model.**    Let $p_\Sigma^\star$ be some probability distribution over $\Sigma^*$ we wish to estimate. As is standard in statistical estimation, we take a large sample $\boldsymbol{\sigma}^{(1)}, \dots, \boldsymbol{\sigma}^{(M)} \overset{\text{i.i.d.}}{\sim} p_\Sigma^\star$. Next, we *encode* each strings of characters into strings of *tokens*, giving us a corpus of token strings: $\boldsymbol{\delta}^{(m)} = \tau(\boldsymbol{\sigma}^{(m)})$ for $1 \leq m \leq M$. Let $p_\Delta^\star$ denote the distribution of these token strings:

$$p_\Delta^\star(\boldsymbol{\delta}) \stackrel{\text{def}}{=} \sum_{\boldsymbol{\sigma} \in \Sigma^*} \mathbb{1}\{\tau(\boldsymbol{\sigma}) = \boldsymbol{\delta}\} p_\Sigma^\star(\boldsymbol{\sigma}) \tag{3}$$

Next, we estimate a token-level language model $p_\Delta$ to fit the strings $\boldsymbol{\delta}^{(1)}, \dots, \boldsymbol{\delta}^{(M)}$. After estimation, we have a distribution $p_\Delta$ over strings of *tokens* rather than strings of *characters*. Thus, to generate a string of characters, we first generate a token string $\boldsymbol{\delta} \sim p_\Delta$, and *decode* it (via $\kappa(\boldsymbol{\delta})$) to get a character string. This process is why the tokenized language model's character strings are distributed according to Eq. (2).[4] Tokenization is used in practice because working with (short) token strings is easier than (long) character strings. The most commonly used tokenization scheme for modern language models is byte-pair encoding (BPE; Sennrich et al., 2016; Gage, 1994), which is a simple and effective algorithm for shortening text (Zouhar et al., 2023).

## 2.5. Byte-Pair Encoding

**Byte-pair encoding** (**BPE**; Gage, 1994; Sennrich et al., 2016) is a commonly used tokenization scheme.[5] Intuitively,

---

[3]Vieira et al. (2024) give algorithms for working with tokenized language models at the character (or byte) level, including algorithms for computing conditional probabilities.

[4]Gastaldi et al. (2025) characterizes the conditions required for tokenized language models to estimate $p_\Sigma^\star$ consistently.

[5]BPE is the most prevalent tokenization strategy in modern language models; it is used by OpenAI's GPT-2 (Radford et al., 2019), GPT-3 (Brown et al., 2020), GPT-3.5 (OpenAI, 2022), and GPT-4 (OpenAI, 2024a), GPT-4o (OpenAI, 2024b); Meta's Llama 1 (Touvron et al., 2023a), Llama 2 (Touvron et al., 2023b), and Llama 3 (Llama Team, 2024); Mistral's 7B (Mistral AI, 2023), Au (Mistral AI, 2024a), and Nemo (Mistral AI, 2024b);

| $\boldsymbol{\delta}$ | $\varphi(\boldsymbol{\delta})$ |
|---|---|
| t  he
83 · 258 | the
1169 |
| t  .
83 · 13 | t  .
83 · 13 |
| t
83 | t
83 |
| th  x
400 · 87 | th  x
400 · 87 |
| th  e
400 · 68 | the
1169 |
| tha  n
12898 · 77 | than
14813 |
| tha  n  ks
12898 · 77 · 591 | thanks
27547 |
| Th  ing
817 · 278 | T  hing
51 · 722 |
| Th  inging
817 · 14146 | T  hing  ing
51 · 722 · 278 |

*Figure 2.* Short examples of canonical and noncanonical token strings from GPT2's tokenizer. Because the canonicalization operation $\varphi$ is idempotent, each example in the second column is canonical. These examples were chosen to highlight some of BPE's unintuitive preferences, e.g., BPE is not a longest-match tokenizer, and canonicalization can *increase* the string's length.

the encoding function of BPE works as follows: each character of an input string $\sigma_1 \cdots \sigma_N$ is initially mapped to an individual token, then pairs of tokens are iteratively merged into bigger tokens according to a list of *merge rules* $\mathcal{M}$. We refer the reader to App. B for a formal definition of BPE.

## 2.6. Canonicality

To facilitate discussion, we start by introducing some terminology. The set of **canonical strings**:

$$\mathcal{D} \stackrel{\text{def}}{=} \{\tau(\boldsymbol{\sigma}) \mid \boldsymbol{\sigma} \in \Sigma^*\} \tag{4a}$$

The set of **noncanonical strings** $\overline{\mathcal{D}} \stackrel{\text{def}}{=} \Delta^* \setminus \mathcal{D}$, and the set of **canonical prefixes**:

$$\overrightarrow{\mathcal{D}} \stackrel{\text{def}}{=} \{\boldsymbol{\delta} \in \Delta^* \colon \boldsymbol{\delta} \preceq \boldsymbol{\delta}', \boldsymbol{\delta}' \in \mathcal{D}\} \tag{4b}$$

We say that $\boldsymbol{\delta}$ is **canonical** if $\boldsymbol{\delta} \in \mathcal{D}$. We say $\boldsymbol{\delta}$ is a **canonical prefix** if $\boldsymbol{\delta} \in \overrightarrow{\mathcal{D}}$. We define the **canonicalization function** $\varphi \colon \Delta^* \to \mathcal{D}$ as $\varphi(\boldsymbol{\delta}) \stackrel{\text{def}}{=} \tau(\kappa(\boldsymbol{\delta}))$. Note that $\mathcal{D} = \{\varphi(\boldsymbol{\delta}) \mid \boldsymbol{\delta} \in \Delta^*\}$. We say that our language model $p_\Delta$ is **canonical** if $\boldsymbol{\delta} \notin \mathcal{D} \implies p_\Delta(\boldsymbol{\delta}) = 0$, i.e., $\sum_{\boldsymbol{\delta} \in \mathcal{D}} p_\Delta(\boldsymbol{\delta}) = 1$.

**Membership tests.**    We can test membership in $\mathcal{D}$ via the **round-trip test**: $\boldsymbol{\delta} \in \mathcal{D} \iff \varphi(\boldsymbol{\delta}) = \boldsymbol{\delta}$. We note, however, that testing membership in $\overrightarrow{\mathcal{D}}$ for general tokenizers is less

---

GPT-NeoX (Black et al., 2022); EleutherAI's Pythia (Biderman et al., 2023); Anthropic's Claude (Anthropic, 2023a) and Claude 2 (Anthropic, 2023b); Google's Gemma (Gemma Team, 2024); and AI2's OLMo 1 (Groeneveld et al., 2024) and OLMo 2 (Team OLMo, 2025).

straightforward.[6]

**Membership tests for BPE.** App. B provides efficient algorithms for membership in these sets for BPE. Lemma 3 shows that for BPE $\mathcal{D} = \overrightarrow{\mathcal{D}}$, i.e., the round-trip test is a correct membership test for $\overrightarrow{\mathcal{D}}$. We also provide a very efficient incremental algorithm for performing these membership tests that exploits the properties of the BPE encoding procedure. Fig. 2 shows examples of canonicalized BPE strings.

## 3. Canonicality by Conditioning

Recall (Eq. (2)) that a *tokenized* language model defines a language model $p_\Sigma(\boldsymbol{\sigma})$ over $\Sigma^*$. However, it is common practice to evaluate the probability of a character string $\boldsymbol{\sigma} \in \Sigma^*$ using the following *sub*-distribution over $\Sigma^*$ during both likelihood-based training and evaluation:

$$p'_\Sigma(\boldsymbol{\sigma}) \overset{\text{def}}{=} p_\Delta(\tau(\boldsymbol{\sigma})) \tag{5}$$

The problem with $p'_\Sigma$ is that $\sum_{\boldsymbol{\sigma} \in \Sigma^*} p'_\Sigma(\boldsymbol{\sigma})$ can be $< 1$, making it is an improper probability distribution over $\Sigma^*$.

Unfortunately, every token string—canonical or not—is assigned a nonzero probability under a softmax-normalized model; thus, modern language models allocate (typically small) probability to each and every noncanonical string. The leakage of mass to noncanonical strings is compounded by the fact that $\mathcal{D}$ is an *exponentially sparse* subset of $\Delta^*$.[7]

Our first approach to this problem defines a language model $g$ that is the result of *probabilistic conditioning* on the event that the generated token string is in $\mathcal{D}$.

**Definition 1.** *The **globally canonicalized language models** $g_\Sigma$ and $g$ are defined as the following probability distributions over $\Sigma^*$ and $\Delta^*$, respectively:*

$$g_\Sigma(\boldsymbol{\sigma}) \overset{\text{def}}{=} g(\tau(\boldsymbol{\sigma})) \tag{6a}$$

$$g(\boldsymbol{\delta}) \overset{\text{def}}{=} \Pr_{Y \sim p_\Delta} [Y = \boldsymbol{\delta} \mid Y \in \mathcal{D}] \tag{6b}$$

$$= \frac{\Pr_{Y \sim p_\Delta}[Y = \boldsymbol{\delta}, Y \in \mathcal{D}]}{\Pr_{Y \sim p_\Delta}[Y \in \mathcal{D}]} \tag{6c}$$

$$= \frac{1}{Z} p_\Delta(\boldsymbol{\delta}) \mathbb{1}\{\boldsymbol{\delta} \in \mathcal{D}\} \tag{6d}$$

*where $Z$ is the **canonicality rate**:*

$$Z \overset{\text{def}}{=} \Pr_{Y \in p_\Delta} [Y \in \mathcal{D}] \tag{6e}$$

---

[6]When the tokenization function $\tau$ is implemented as a finite-state transducer, we may derive an efficient finite-state automaton that describes the prefix language of its outputs.

[7]More precisely, as a function of a length limit $N$, the fraction of canonical strings in the universal of token strings $\frac{|\{\boldsymbol{\delta} \in \mathcal{D} : |\boldsymbol{\delta}| \leq N\}|}{|\{\boldsymbol{\delta} \in \Delta^* : |\boldsymbol{\delta}| \leq N\}|}$ decreases exponentially quickly. This is evident as each character string has one canonical encoding, and exponentially many noncanonical encodings as a function of its length.

Note that $g_\Sigma(\boldsymbol{\sigma}) = \frac{1}{Z} p_\Delta(\tau(\boldsymbol{\sigma}))$, meaning that we may interpret the globally canonicalized model as renormalizing $p'_\Sigma$. We note that the effect of conditioning the language model to generate only canonical token strings may dramatically change the conditional prefix distributions of the distribution. We will discuss the details of how to approximately and exactly generate samples from $g$.

### 3.1. Theoretical Guarantees for Conditioning

**Why do we condition?** The conditional distribution $g$ lets us preserve what matters: the relative probabilities of canonical strings, ensuring that we eliminate the wasteful mass allocated to noncanonical strings without distorting the existing mass on canonical strings.[8]

The following proposition shows that test-time conditioning improves our estimate $g$ of the true distribution $p_\Delta^\star$.

**Proposition 1.** *Assuming that the true distribution over tokens $p_\Delta^\star$ is canonical, the globally canonicalized model $g$ guarantees the following reduction in KL divergence:*

$$\text{KL}(p_\Delta^\star \| p_\Delta) - \text{KL}(p_\Delta^\star \| g) = \underbrace{-\log Z}_{\geq 0} \tag{7}$$

Proof: See App. D.

We may interpret Proposition 1 as saying that, no matter how good or how bad of a model the original language model $p_\Delta$ is with respect to the true distribution $p_\Delta^\star$, we can expect a reduction in the KL divergence equal to $-\log Z$, bringing it *closer* to $p_\Delta^\star$. This reduction is large if $Z$ is small, i.e., the token strings generated in $p_\Delta$ are unlikely to be canonical, modest if $Z \approx 1$, and zero if $Z = 1$. In other words, the change in KL quantifies the value of having the constraint as prior knowledge.

### 3.2. Algorithms for Approximate Conditioning

#### 3.2.1. EXACT ALGORITHMS

The simplest exact conditioning algorithm for sampling from $g$ is **rejection sampling**:

```
1 def rejection_sampling():
2     while True:
3         δ ∼ p_Δ
4         if δ ∈ D: return δ
```

However, the rejection sampling algorithm can be inefficient if $Z$ is small, as its expected running time per sample is $\mathcal{O}(1/Z)$. When $Z \approx 1$, this is pretty fast. However, when it is smaller, it can be unusably slow.

---

[8]More formally, the conditional distribution is the *unique* distribution $q$ over $\mathcal{D}$ that maintains the relative probabilities of any pair $\boldsymbol{\delta}, \boldsymbol{\delta}' \in \mathcal{D}$—that is, $p_\Delta(\boldsymbol{\delta}'), q(\boldsymbol{\delta}) = p_\Delta(\boldsymbol{\delta}), q(\boldsymbol{\delta}')$. The only way to achieve this is to set $q(\boldsymbol{\delta}) = \eta, p_\Delta(\boldsymbol{\delta})$ for all $\boldsymbol{\delta} \in \mathcal{D}$, where normalization fixes $\eta = 1/Z$. Thus, $q = g$ is the unique way to preserve relative probability subject to the canonicality constraint.

Another correct sampling algorithm is **ancestral sampling**:

```
5  def ancestral_sampling():
6      δ ← ε
7      while True:
8          δ' ∼ g⃗(· | δ)
9          if δ' = EOS: return δ
10         δ ← δ·δ'
```

Here, each step of the algorithm samples a token (or EOS) from the globally canonicalized model's conditional prefix distribution. Unfortunately, computing $\overrightarrow{g}$ exactly is intractable as it requires summing over infinitely many future strings. Thus, $\overrightarrow{g}$ must be approximated.[9] In §3.2.2, we will provide a cheap, local approximation and, in §3.2.3, we will devise a strategy for improving the short-sightedness of the local approximation by using more computation.

### 3.2.2. A LOCALLY CANONICALIZED APPROXIMATION

Our locally canonicalized model $\ell$ is a distribution over $\Delta^*$ that approximates `ancestral_sampling` for sampling from $g$ by using the following local approximation $\overrightarrow{\ell}$ to the global prefix probability $\overrightarrow{g}$.

**Definition 2.** *The **locally canonicalized model** $\ell$ is a language model over $\Delta^*$, which assigns probability zero to noncanonical strings. The probability of a token string under $\ell$ is defined auto-regressively as*

$$\ell(\boldsymbol{\delta}) \stackrel{\text{def}}{=} \overrightarrow{\ell}(\text{EOS} \mid \boldsymbol{\delta}) \prod_{t=1}^{|\boldsymbol{\delta}|} \overrightarrow{\ell}(\delta_t \mid \boldsymbol{\delta}_{<t}) \qquad (9a)$$

*where its next-token conditional distributions are defined as*

$$\overrightarrow{\ell}(\text{EOS} \mid \boldsymbol{\delta}) \stackrel{\text{def}}{=} \frac{\overrightarrow{p_\Delta}(\text{EOS} \mid \boldsymbol{\delta}) \mathbb{1}\{\boldsymbol{\delta} \in \mathcal{D}\}}{\overrightarrow{\ell}(\boldsymbol{\delta})} \qquad (9b)$$

$$\overrightarrow{\ell}(\delta' \mid \boldsymbol{\delta}) \stackrel{\text{def}}{=} \frac{\overrightarrow{p_\Delta}(\delta' \mid \boldsymbol{\delta}) \mathbb{1}\{\boldsymbol{\delta}·\delta' \in \overrightarrow{\mathcal{D}}\}}{\overrightarrow{\ell}(\boldsymbol{\delta})} \qquad (9c)$$

*where $\overrightarrow{\ell}(\boldsymbol{\delta})$ normalizes the distribution over $\Delta \cup \{\text{EOS}\}$ that lead to a prefix-canonical extension of the string:*

$$\overrightarrow{\ell}(\boldsymbol{\delta}) \stackrel{\text{def}}{=} \mathbb{1}\{\boldsymbol{\delta} \in \mathcal{D}\} \overrightarrow{p_\Delta}(\text{EOS} \mid \boldsymbol{\delta}) \qquad (9d)$$
$$+ \sum_{\delta' \in \Delta} \overrightarrow{p_\Delta}(\delta' \mid \boldsymbol{\delta}) \mathbb{1}\{\boldsymbol{\delta}·\delta' \in \overrightarrow{\mathcal{D}}\}$$

This distribution is practical for the following reasons:

1. It is efficient: The canonicality checks are cheap to compute (App. B), and there is very little additional overhead over sampling from the base conditional distribution.

2. It has the correct support: The local distribution can only place positive probability on canonical strings; it eliminates noncanonical strings by masking out any token $\delta'$ that would render the extended string $\boldsymbol{\delta}·\delta'$ noncanonical.

3. It often gives a reasonable approximation to the conditional global distribution: We expect this to be a good approximation when $\overrightarrow{\ell}(\boldsymbol{\delta}) \approx 1$, which we may expect in practice as noncanonical strings tend to be assigned tiny probabilities, making the sum in Eq. (9d) close to 1.

Unfortunately, the locally canonicalized model $\ell$ gives a *warped* approximation to $g$, as Proposition 2 shows:

**Proposition 2.**

$$\ell(\boldsymbol{\delta}) = p_\Delta(\boldsymbol{\delta}) \mathbb{1}\{\boldsymbol{\delta} \in \mathcal{D}\} \underbrace{\left[ \prod_{t=1}^{|\boldsymbol{\delta}|+1} \overrightarrow{\ell}(\boldsymbol{\delta}_{<t}) \right]^{-1}}_{\stackrel{\text{def}}{=} w_\ell(\boldsymbol{\delta})} \qquad (10a)$$

*Thus,*

$$w_\ell(\boldsymbol{\delta}) = \frac{p_\Delta(\boldsymbol{\delta})}{\ell(\boldsymbol{\delta})} \mathbb{1}\{\boldsymbol{\delta} \in \mathcal{D}\} \qquad (10b)$$

Proof: See App. D.

We call $w_\ell(\boldsymbol{\delta})$ the **weight** of the locally canonicalized sample (in relation to $p_\Delta^\star$). We note that the weight describes the change in relative probability of the canonical token strings $\boldsymbol{\delta} \in \mathcal{D}$. We call this unwanted change **warping**. There is no warping in the globally canonicalized case, as the relative probability is unchanged. In §3.2.3, we describe methods that can *un*warp the local distribution at the cost of additional computation.

Despite the warping, the local distribution $\ell$ comes with the following reassuring theoretical property, guaranteeing that it helps estimate the true distribution $p_\Delta^\star$.

**Proposition 3.** *Assuming that the true distribution over tokens $p_\Delta^\star$ is canonical, the locally canonicalized model $\ell$ guarantees the following reduction in KL divergence:*

$$\text{KL}(p_\Delta^\star \,\|\, p_\Delta) - \text{KL}(p_\Delta^\star \,\|\, \ell) = \underbrace{- \mathop{\mathbb{E}}_{\boldsymbol{\delta} \sim p_\Delta^\star} [\log w_\ell(\boldsymbol{\delta})]}_{\geq 0} \quad (11)$$

Proof: See App. D.

Proposition 3 confirms the intuitive notion that local canonicalization cannot make the language model *worse*.[10]

### 3.2.3. SAMPLING-BASED INFERENCE

Unfortunately, sampling from $g$ is hard because we cannot efficiently compute the conditional prefix probability $\overrightarrow{g}$

---

[9]We also mention more sophisticated approximations, such as learning (Zhao et al., 2024), adaptive upper bounds (Park et al., 2025), and sequential Monte Carlo steering (Lew et al., 2023; Loula et al., 2025). We do not explore these more sophisticated strategies, as the local approximation works surprisingly well.

[10]Proposition 3 may be generalized to any distribution $q$. Specifically, $\text{KL}(p_\Delta^\star \,\|\, p_\Delta) - \text{KL}(p_\Delta^\star \,\|\, q) = -\mathbb{E}_{\boldsymbol{\delta} \sim p_\Delta^\star}\left[\log \frac{p_\Delta(\boldsymbol{\delta})}{q(\boldsymbol{\delta})}\right]$. However, it may not necessarily be a *reduction*, as it is for $\ell$, if the $q$ distorts the distribution too much.

(and rejection sampling is slow when $Z$ is small). In this section, we show how to use the local approximation in conjunction with sampling-based inference to improve the fidelity to $g$ at the cost of increased computation. These methods use the local distributions as a *proposal* distribution to produce candidates that will be properly weighted and resampled in a manner so that they maintain a principled approximation to the global distribution $g$. Therefore, in expectation and in the limit, these methods produce exact samples; thus, they do not warp the distribution.

The warping in the locally canonicalized method can occur because the sampling algorithm approximated the conditional prefix probability canonicality, meaning that we may sample a string of tokens that looks good initially, but we end up stuck with a bad string prefix because we overestimated the conditional prefix probability. In contrast, rejection sampling *restarts* the sample once noncanonicality is detected, whereas the local method *ignores* noncanonical tokens and *forces* the current string to complete. We can fix the bias in the local approximation by exploiting our knowledge of $w_\ell(\boldsymbol{\delta})$ for each sample $\boldsymbol{\delta} \sim \ell$. More specifically, $w_\ell(\boldsymbol{\delta})$ is proportional to the over- or under-representation rate of $\boldsymbol{\delta}$ in samples from $\ell$ with respect to the target distribution $g$. Therefore, we can correctly account for this *in expectation* by *weighting* each sample by its $w_\ell$ factor. This is the principle behind the importance sampling methods we describe below.[11]

**Proposition 4.**

$$Z = \mathop{\mathbb{E}}_{\boldsymbol{\delta} \sim \ell}[w_\ell(\boldsymbol{\delta})] = \mathop{\mathbb{E}}_{\boldsymbol{\delta} \sim \ell}\left[\frac{p_\Delta(\boldsymbol{\delta})\mathbb{1}\{\boldsymbol{\delta} \in \mathcal{D}\}}{\ell(\boldsymbol{\delta})}\right] \quad (12)$$

Proof: See App. D.

Thus, we can apply **importance sampling** to obtain an unbiased estimate $\widehat{Z}$ of $Z$ given $\underline{\boldsymbol{\delta}}^{(1)}, \dots, \underline{\boldsymbol{\delta}}^{(M)} \stackrel{\text{i.i.d.}}{\sim} \ell$:

$$\widehat{Z} \stackrel{\text{def}}{=} \frac{1}{M}\sum_{m=1}^{M} w_\ell(\underline{\boldsymbol{\delta}}^{(m)}) \quad (13)$$

We may also estimate the global distribution $g$ as follows:

$$\widehat{g}(\boldsymbol{\delta}) \stackrel{\text{def}}{=} \frac{1}{M \cdot \widehat{Z}}\sum_{m=1}^{M} w_\ell(\underline{\boldsymbol{\delta}}^{(m)})\mathbb{1}\{\boldsymbol{\delta} = \underline{\boldsymbol{\delta}}^{(m)}\} \quad (14)$$

Note that the quotient with $\widehat{Z}$ leads to statistical bias; however, that bias vanishes as $M$ grows larger. If desired, we may also use $\widehat{g}$ for drawing approximate samples using a method known as **importance resampling**. This method produces approximate samples from $g$ by sampling $\boldsymbol{\delta} \sim \widehat{g}$,

which has a support set with size at most $M$. Thus, importance sampling and resampling give us a parameter $M$ that improves the approximation as we make it larger.[12]

## 4. Canonicality by Construction

The method we present in this section augments the output layer of the transformer architecture with canonicality constraints. Thus, it requires additional training and test-time computation. Fortunately, this additional computation, at least in the case of BPE, is minimal thanks to efficient algorithms for checking canonicality (see App. B).

Suppose we have a **base architecture**, i.e., a parametric family of language models $\{p_{\boldsymbol{\theta}}\}_{\boldsymbol{\theta} \in \mathbb{R}^D}$ that is parameterized by a real-valued vector of parameters $\boldsymbol{\theta} \in \mathbb{R}^D$. We assume $\log p_{\boldsymbol{\theta}}(\boldsymbol{\delta})$ is continuously differentiable with respect to $\boldsymbol{\theta}$ for all $\boldsymbol{\delta} \in \Delta^*$, and that each $p_{\boldsymbol{\theta}}$ is a valid language model over $\Delta^*$. For example, $\{p_{\boldsymbol{\theta}}\}_{\boldsymbol{\theta} \in \mathbb{R}^D}$ can be a parametric family of transformers or recurrent neural networks.[13] Let $\overrightarrow{p_{\boldsymbol{\theta}}}$ denote the conditional prefix probability of one such model.

**Definition 3.** *Our **canonicalized architecture** is a parametric family of language models $\{\ell_{\boldsymbol{\theta}}\}_{\boldsymbol{\theta} \in \mathbb{R}^D}$ that is built on top of a base architecture $\{p_{\boldsymbol{\theta}}\}_{\boldsymbol{\theta} \in \mathbb{R}^D}$. We define each $\ell_{\boldsymbol{\theta}}$ in terms of $p_{\boldsymbol{\theta}}$ in the following manner:*

$$\ell_{\boldsymbol{\theta}}(\boldsymbol{\delta}) \stackrel{\text{def}}{=} \overrightarrow{\ell_{\boldsymbol{\theta}}}(\text{EOS} \mid \boldsymbol{\delta}) \prod_{t=1}^{|\boldsymbol{\delta}|} \overrightarrow{\ell_{\boldsymbol{\theta}}}(\delta_t \mid \boldsymbol{\delta}_{<t}) \quad (15a)$$

*where each $\overrightarrow{\ell_{\boldsymbol{\theta}}}(\cdot \mid \boldsymbol{\delta}_{<t})$ is defined as one of the following distributions over $\Delta \cup \{\text{EOS}\}$:*

$$\overrightarrow{\ell_{\boldsymbol{\theta}}}(\delta' \mid \boldsymbol{\delta}) \stackrel{\text{def}}{=} \frac{\overrightarrow{p_{\boldsymbol{\theta}}}(\delta' \mid \boldsymbol{\delta})\,\mathbb{1}\{\boldsymbol{\delta}\cdot\delta' \in \overrightarrow{\mathcal{D}}\}}{\overrightarrow{\ell_{\boldsymbol{\theta}}}(\boldsymbol{\delta})} \quad (15b)$$

$$\overrightarrow{\ell_{\boldsymbol{\theta}}}(\text{EOS} \mid \boldsymbol{\delta}) \stackrel{\text{def}}{=} \frac{\overrightarrow{p_{\boldsymbol{\theta}}}(\text{EOS} \mid \boldsymbol{\delta})\,\mathbb{1}\{\boldsymbol{\delta} \in \mathcal{D}\}}{\overrightarrow{\ell_{\boldsymbol{\theta}}}(\boldsymbol{\delta})} \quad (15c)$$

*and $\overrightarrow{\ell_{\boldsymbol{\theta}}}(\boldsymbol{\delta})$ ensures normalization:*

$$\overrightarrow{\ell_{\boldsymbol{\theta}}}(\boldsymbol{\delta}) \stackrel{\text{def}}{=} \overrightarrow{p_{\boldsymbol{\theta}}}(\text{EOS} \mid \boldsymbol{\delta})\,\mathbb{1}\{\boldsymbol{\delta} \in \mathcal{D}\} \quad (15d)$$
$$+ \sum_{\delta''} \overrightarrow{p_{\boldsymbol{\theta}}}(\delta'' \mid \boldsymbol{\delta})\,\mathbb{1}\{\boldsymbol{\delta}\cdot\delta'' \in \overrightarrow{\mathcal{D}}\}$$

*Much like the locally canonicalized model (Def. 2), non-canonical strings are forced to have probability zero using canonicality checks in the definition of its conditional prefix probabilities Eq. (15b) and (15c).*

---

[11]For a rigorous treatment of the technical conditions required for importance resampling and estimation to converge, we refer the reader to Chatterjee & Diaconis (2017).

[12]We note that for these estimates to converge to their correct values, we require $p_\Delta(\boldsymbol{\delta})\mathbb{1}\{\boldsymbol{\delta} \in \mathcal{D}\} > 0 \implies \ell(\boldsymbol{\delta}) > 0$ for all $\boldsymbol{\delta} \in \Delta^*$. This is ensured when our prefix canonicality test $\mathbb{1}\{\boldsymbol{\delta} \in \overrightarrow{\mathcal{D}}\}$ is *exact*. More generally, it is safe to allow for false positives but not false negatives.

[13]Our experiments use transformer-based language models.

**Why train?** Suppose the original model $p_\triangle$ is equal to some $p_\theta$ in our family; let $\theta^{(0)}$ denote its parameters. Clearly, if we set $\theta = \theta^{(0)}$, the model $\ell_\theta$ is no different from $\ell$. The reason we optimize $\theta$ beyond $\theta^{(0)}$ is that the training objective for $\theta^{(0)}$ pressured the parameters to model canonicality, but now that pressure is gone because the canonicalized architecture (Def. 3) enforces that constraint for all $\theta$. Thus, the parameters that were previously used to model canonicality preferences can be repurposed to model any other textual phenomena in the training data.

**Training $p_\theta$.** Given a training corpus $\sigma^{(1)}, \ldots, \sigma^{(M)} \overset{\text{i.i.d.}}{\sim} p_\Sigma^\star$ of character strings, let $\delta^{(1)}, \ldots, \delta^{(M)}$ be their corresponding (canonical) tokenizations. We define $\mathcal{L}(p)$, the **log-loss** (average negative log-likelihood) of the training corpus under a language model $p$:

$$\mathcal{L}(p) \overset{\text{def}}{=} -\frac{1}{M} \sum_{m=1}^{M} \log p(\delta^{(m)}) \qquad (16)$$

Log-loss is a reasonable (and common) training objective, as minimizing $\mathcal{L}(\ell_\theta)$ also minimizes $\text{KL}(p_\Sigma^\star \| \ell_\theta)$ in the limit of infinite data (with perfect optimization). However, training a large language model with $\mathcal{L}$ from scratch is prohibitively costly. Instead, we fine-tune an existing (non-canonical) large language model.

We use the following **fine-tuning objective**, which strikes a balance between fitting the log-loss objective while maintaining some fidelity to the original language model $p_\triangle$:

$$\mathcal{F}_\lambda(\theta) \overset{\text{def}}{=} (1 - \lambda) \cdot \mathcal{L}(\ell_\theta) + \lambda \cdot \text{KL}(\ell_\theta \| p_\triangle) \qquad (17)$$

Here, we use the KL divergence between $\ell_\theta$ and the original unconstrained model distribution $p_\triangle$ as a regularizer (e.g., Christiano et al., 2017; Stiennon et al., 2020; Ziegler et al., 2020; Korbak et al., 2022). The **regularization parameter** $0 \le \lambda \le 1$ is used to trade fidelity to the original model (higher) against fidelity to the fine-tuning data (lower).

**Optimization.** Our optimization algorithm is based on stochastic gradient descent. To optimize $\mathcal{F}_\lambda(\theta)$, on each step, we randomly choose with probability $\lambda$ which of the two terms to perform a gradient step on.

1. For the log-loss updates, we sample a minibatch of examples from the fine-tuning corpus and use the sample average of that minibatch's gradient as an efficient approximation to the gradient over the entire corpus.
2. We estimate the gradient of the KL regularization term using Amini et al.'s (2025) Rao–Blackwellized method using samples from the current $\ell_\theta$.

Experiments initialize $\ell_\theta$ with the same parameters as $p_\triangle$.

# 5. Experiments

This section evaluates our proposed methods—*canonicality by constraints* (global and local; §3) and *canonicality by conditioning* (§4)—by measuring their impact on real datasets and language models. This evaluation complements the theoretical guarantees of Proposition 1 and Proposition 3, by quantifying the log-loss ($\mathcal{L}$; Eq. (16)) on held-out data.

## 5.1. Setup

**Datasets.** We experiment on the following two datasets:

- **Penn Treebank** (**PTB**, Marcus et al., 1993) (test split; 3761 strings, 82k words, 439k characters)
- **WikiText** (Merity et al., 2017) (test split; 4358 strings, 234k words and 1286k characters)

**Models.** We experiment with the following models:

- **GPT-2 models** (Radford et al., 2019) of increasing size (small, medium, and large)
- **Llama models** (Llama Team, 2024) of increasing size (**3.2-1B**, **3.2-3B**, and **3.1-8B**).

**Pre-tokenization.** Our efficient bigram-based canonicality test $\mathbb{1}\{\delta \cdot \delta' \in \overrightarrow{\mathcal{D}}\}$ (App. B) makes the simplifying assumption that the tokenizer $\tau$ is only based on BPE. However, in practice, systems use an additional **pre-tokenization** step that breaks text into chunks called pre-tokens (e.g., based on spaces and punctuation). Most pre-tokenizers are based on hand-crafted regular expressions; the GPT-2 and Llama models each use a distinct pre-tokenization regular expressions. Since these models make use pre-tokenization, our incremental test occasionally makes some mistakes. Particularly, concerning are instances of *false negatives*, i.e., we rule out a bigram as noncanonical when it is in fact canonical. False negatives are problematic for estimation (Footnote 12). These errors are caused by the interaction of the pre-tokenizer and the tokenizer.[14] To work around these rare exceptions, we determined a (small) set of *overrides* for our canonicality test by identifying the bigrams where we have made a false negative judgment on a corpus of canon-

---

[14] For example, under the GPT-2 tokenizer, the token bigram $\overset{\text{\n}}{198} \cdot \overset{\text{\n}}{198}$ may or may not be canonical, depending on the context. For instance, in the following example, it is canonical:

$$\tau(\text{Hi,\textbackslash n\textbackslash nI}) = \overset{\text{Hi}}{17250} \cdot \overset{,}{11} \cdot \overset{\text{\n}}{198} \cdot \overset{\text{\n}}{198} \cdot \overset{\text{I}}{40}$$

because the pre-tokenizer creates separate pretokens for each newline, preventing BPE from merging them. However, in the next example, it is not canonical:

$$\tau(\text{Hi,\textbackslash n\textbackslash n}) = \overset{\text{Hi}}{17250} \cdot \overset{,}{11} \cdot \overset{\text{\textbackslash n\textbackslash n}}{628}$$

because the pre-tokenizer does not create separate pretokens for the newlines, allowing BPE to merge them.

ical strings. We note that this workaround occasionally introduces false positives.[15]

## 5.2. Canonicality by Conditioning

**Methology.** Fig. 3 reports the log-loss of the local $\ell$ and global $g$ methods on each dataset ($\{p_\Delta^{(m)}\}_{m=1}^M$) as well as $p_\Delta$, which serves as our baseline. Below are the details of how we estimated the log-loss of each method.

- *Baseline*: The log-loss for the baseline method is

$$\mathcal{L}(p_\Delta) = -\frac{1}{M}\sum_{m=1}^M \log p_\Delta(\boldsymbol{\delta}^{(m)}) \qquad (18a)$$

- *Global*: The log-loss for the global method is

$$\mathcal{L}(g) = \mathcal{L}(p_\Delta) + \log Z \approx \mathcal{L}(p_\Delta) + \log \widehat{Z} \qquad (18b)$$

where $\widehat{Z}$ is computed by Eq. (13) using 2000 samples from $\ell$. Note that $\log \widehat{Z} \leq 0$; thus, the global method can only improve the log-loss, $\mathcal{L}(g) \leq \mathcal{L}(p_\Delta)$.[16]

- *Local*: The log-loss for the local method is

$$\mathcal{L}(\ell) = \mathcal{L}(p_\Delta) + \underbrace{\frac{1}{M}\sum_{m=1}^M \log w_\ell(\boldsymbol{\delta}^{(m)})}_{\overset{\text{def}}{=}\widehat{W}} \qquad (18c)$$

Note that $\frac{1}{M}\sum_{m=1}^M \log w_\ell(\boldsymbol{\delta}^{(m)}) \leq 0$; thus, the local method can only improve the log-loss, $\mathcal{L}(\ell) \leq \mathcal{L}(p_\Delta)$.

**Relationship to theory.** Proposition 1 shows that the reduction in KL divergence is equal to $-\log Z$, which is equal to the difference in log-loss $\mathcal{L}(p_\Delta) - \mathcal{L}(g)$. Proposition 3 shows that the reduction in KL divergence is equal to $-\mathbb{E}_{\boldsymbol{\delta}\sim p_\Delta^\star}[\log w_\ell(\boldsymbol{\delta})]$, which is equal to the difference in log-loss $\mathcal{L}(p_\Delta) - \mathcal{L}(\ell)$ (in expectation).

**Observations.**

- For the global method, the change in the log-loss is *independent* of the dataset because $\log \widehat{Z}$ is *independent* of the dataset. However, for the local method, it is *dependent* because $\widehat{W}$ is *dependent* on the dataset.
- Statistical significance:
  - For the global method, the log-loss reduction is *positive* and *constant* for every string; therefore, it is trivially statistically significantly better than the baseline method according to the paired-permutation test (i.e., $p = 0$).

|  |  | Model | Baseline | Local | Global |
|---|---|---|---|---|---|
| PTB | GPT-2 | small | 201.0 | 200.7 | **199.1** |
|  |  | medium | 195.1 | 194.5 | **193.1** |
|  |  | large | 189.4 | 188.9 | **188.2** |
|  | Llama | 1B | 171.2 | 171.1 | **169.7** |
|  |  | 3B | 165.0 | 165.0 | **164.2** |
|  |  | 8B | 161.5 | 161.5 | **160.1** |
| WikiText | GPT-2 | small | 369.2 | **367.0** | 367.3 |
|  |  | medium | 334.1 | 333.2 | **332.2** |
|  |  | large | 320.8 | **319.1** | 319.6 |
|  | Llama | 1B | 286.7 | **284.4** | 285.2 |
|  |  | 3B | 264.6 | **262.0** | 263.7 |
|  |  | 8B | 248.2 | **245.8** | 246.8 |

*Figure 3.* Log-loss ($\mathcal{L}$; bits/string) for the baseline ($p_\Delta$), local ($\ell$), and global ($g$) methods across two datasets and models. Bolding indicates that the number is the best in its row. See text for discussion of statistical significance.

- For the local method, the log-loss reduction is *positive* for all strings, but the amount *varies* across strings; nonetheless, it is trivially statistically significantly better than the baseline method according to the paired-permutation test (i.e., $p = 0$).
- Local vs. global: On the PTB dataset, we found that the global method is significantly better than the local method for all models. However, on the WikiText dataset, the local model is better in all cases except GPT-2-medium.

**Takeways.** On principle (§3.1), we maintain that the global method (based on conditioning) is the correct solution, as it is the only distribution that preserves the relative probabilities for canonical strings. Additionally, our experiments support the local method as a practical approximation to the global distribution, which we expected to be the case because the probability allocated to each noncanonical token is generally quite small, meaning that the warping is not large; see discussion Item 3 in §3.2.2). Interestingly, on some datasets (e.g., WikiText), the warping induced by the local method with respect to the global distribution can be advantageous, i.e., its reduction in log-loss with respect to the WikiText distribution is larger than that of the global method.

**Log-canonicality rate vs. length.** We found that the GPT-2 models generate very long strings, so we investigated the tradeoffs in the log-canonicality rate for truncating the lengths of the strings generated by each model. Fig. 4 shows the effect of length on the log-canonicality rate estimate $\log \widehat{Z}$, using 2000 samples taken from $\ell$. We see that for the Llama methods, the log-canonicality rate stabilizes after 256 tokens, but for the GPT-2 models, it continues decreasing. Thus, for practical reasons, our experiments with GPT-2

---

[15]A complete solution to the pre-tokenization challenges would build a transducer that accurately models the pre-tokenizer, which we then compose with a transducer implementing BPE (see App. B.4). We leave this investigation for future work.

[16]In the case of the GPT-2 models, we restricted strings to have length $\leq 1024$, as these models tend to ramble. We analyze this choice further in Fig. 4.

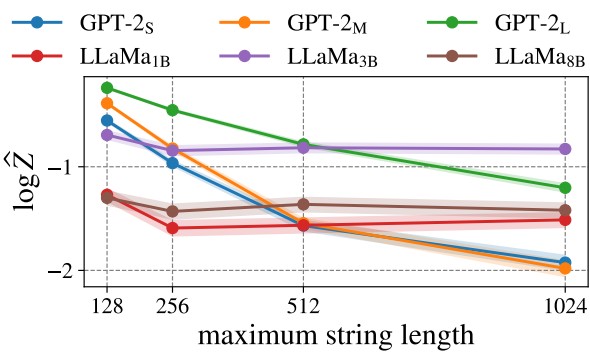

*Figure 4.* Log-canonicality rate vs. (tokenized) length, including 95% confidence intervals, for each model.

| Data | Model | Method | N/A | .001 | .01 | .1 | .2 |
|------|-------|--------|-----|------|-----|----|----|
| PTB | S | $p$ | 201.4 | 143.7 | 143.7 | 143.6 | 146.8 |
| | | $\ell$ | **200.7** | **143.6** | **143.6** | 143.6 | 146.8 |
| | M | $p$ | 195.0 | 128.6 | 128.9 | 128.6 | 132.1 |
| | | $\ell$ | **194.5** | **128.3** | **128.7** | **128.4** | **132.0** |
| WikiText | S | $p$ | 369.2 | 331.7 | 331.7 | 331.7 | 335.7 |
| | | $\ell$ | **367.0** | 331.7 | 331.7 | 331.7 | **335.6** |
| | M | $p$ | 334.1 | 293.9 | 294.0 | 293.9 | **298.7** |
| | | $\ell$ | **333.2** | **293.7** | **293.8** | **293.8** | 298.8 |

*Figure 5.* Log-loss (bits/string) of fine-tuned models on held-out portions of WikiText and PTB. The column labeled N/A reports the log-loss of the baselines $p_\triangle$ and $\ell$. The other columns correspond to the value of the regularization parameter $\lambda$ used in the fine-tuning loss. The rows labeled $p$ are the original architecture and $\ell$ canonicalize architecture.

will be based on a limit length of $1024$, but bear in mind that this means that the actual log-canonicality rate is likely to be much smaller, meaning the reduction in log-loss for GPT-2 is larger than Fig. 3 indicates.

**Supplementary error analysis.** App. E analyzes the most frequent canonicality errors made by each model.

### 5.3. Canonicality by Construction

**Methodology.** We fine-tuned two language models, GPT-$2_S$ and GPT-$2_M$,[17] on the PTB train set and a subset of the WikiText train set with 50K strings and 4.2M words. We consider fine-tuning the canonicalized architecture ($\ell_{\boldsymbol{\theta}}$) and the original architecture ($p_{\boldsymbol{\theta}'}$) using the training criterion $\mathcal{F}_\lambda$ for $\lambda \in \{0.001, 0.01, 0.1, 0.2\}$.[18] As a baseline, we consider the original model $p_\triangle$ and the locally constrained model $\ell$, as these models serve as initialization for fine tuning. Fig. 5 reports the log-loss $\mathcal{L}$ (bits/string) on the held-out datasets for each method mentioned above.

**Observations.**

- In all cases (i.e., models, datasets, and methods), we observe a large improvement in log-loss from fine-tuning. The reduction in log-loss is larger in the case of PTB.
- Provided that $\lambda$ is small enough, we see a consistent reduction in log-loss when comparing $\ell_{\boldsymbol{\theta}}$ and $p_{\boldsymbol{\theta}'}$. However, the difference between $\ell_{\boldsymbol{\theta}}$ and $p_{\boldsymbol{\theta}'}$ is comparatively smaller than we saw in Fig. 3.
- Generally, a the smaller values of $\lambda$ work best, indicating

---

[17]Unfortunately, we are unable to fine-tune larger models due to computational constraints. However, the models used here serve as a proof of concept.

[18]Each model is trained for 3 epochs using the AdamW optimizer (Loshchilov & Hutter, 2019) with a learning rate of $5\mathrm{e}^{-5}$ and linear learning rate decay. For efficiency, we use `bfloat16` to represent the model parameters. We use a minibatch of size 8 for estimating the gradient of each term of the $\mathcal{F}_\lambda$ objective.

that reguarlization towards the original model should not be done too strongly.

**Takeaways.** Unsurprisingly, fine-tuning for the specific dataset is useful regardless of whether we use the original or canonicalized architecture. Fine-tuning with the canonicalized architecture performs slightly better, but improvements appear to be small. It is possible that training the canonicalized architecture from scratch on a huge dataset would yield better results than our proof-of-concept experiment.

## Conclusion

We have demonstrated that enforcing canonicality in token-level language models eliminates systemic probability mass misallocation, leading to improved likelihoods on held-out data across multiple models and corpora. Our proposed methods—canonicality by conditioning and canonicality by construction—provide practical solutions that either refine inference or modify the model architecture to ensure only canonical token strings are assigned positive probability. In addition to the empirical benefits, our theoretical results establish that correcting these mistakes strictly improves model fit. Moreover, our discovery of an efficient incremental test for BPE canonicality simplifies prior approaches, making it more accessible for practical deployment. These findings underscore the importance of aligning token-based probability models with their training distributions, thereby paving the way for more accurate language modeling.

## Acknowledgments

The authors would like to thank John Terilla, Marco Cognetta, Ben Lipkin, Luca Malagutti, and Manuel de Prada Corral for their helpful feedback and discussions.

## Impact Statement

**Potential benefits.** By introducing methods to enforce canonicality, we improve a given model's fidelity to the underlying distribution that generated the text. These contributions have broad implications for the reliability of language models, particularly in applications where precise probability estimates are crucial. By eliminating avoidable modeling errors, we advance the goal of more reliable and robust language models.

**Potential harms.** Constraining tokenization may inadvertently reduce robustness to rare, noisy, or adversarial inputs by ruling out alternative tokenizations of the same character string. While the canonicality constraint does not remove any character strings from the language, it does restrict the model to assign nonzero probability only to a single canonical tokenization of each string. This may disproportionately affect rare or unconventional spellings, as their probability mass may be less sharply concentrated on a single tokenization. Future work should provide a deeper analysis of whether or not canonicality enforcement helps or hurts in these settings to ensure fair and robust language model behavior across diverse linguistic contexts.

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

# A. Limitations and Future Work

**Future experimental work.**    A limitation of our experimental design is that we only evaluate our approach's ability to estimate probabilities. We do not have any experiments evaluating any qualitative difference in the sample quality from the canonicalized methods.

**Models overregularize.**    Language models often place a higher probability on a noncanonical encoding of the string than the canonical one. Here is an interesting example from WikiText corpus (Merity et al., 2017):

```
11 xs = """\n\n = Robert Boulter = \n\n\n\n\n Robert Boulter is an English film , \
12 television and theatre actor . He had a guest @-@ starring role on the television \
13 series The Bill in 2000 . """.encode("utf-8")
```

In the case of GPT-2, the conditional probability of the canonical tokenization is only 2.7%, and it is ranked 4[th] in the (partially enumerated) set of encodings of this string:

0.7799: \n\n _= _Robert _B oul ter _= _ _ \n _\n _\n\n _\n _Robert _B oul ter _is _an _English _film _, _television _and _theatre
628 796 5199 347 2852 353 796 220 198 198 628 198 5199 347 2852 353 318 281 3594 2646 837 5581 290 21421
_actor _. . _He _had _a _guest _ _ @ - @ _starring _role _on _the _television _series _The _Bill _in _2000 _. . _
8674 764 679 550 257 8319 220 31 12 31 20495 2597 319 262 5581 2168 383 3941 287 4751 764 220

0.1151: \n\n _= _Robert _B oul ter _= _ _ \n _\n _\n\n _\n _Robert _B oul ter _is _an _English _film _, _television _and _theatre
628 796 5199 347 2852 353 796 220 198 198 628 198 5199 347 2852 353 318 281 3594 2646 837 5581 290 21421
_actor _. . _He _had _a _guest _@ - @ _starring _role _on _the _television _series _The _Bill _in _2000 _. . _
8674 764 679 550 257 8319 2488 12 31 20495 2597 319 262 5581 2168 383 3941 287 4751 764 220

0.0493: \n\n _= _Robert _B oul ter _= _ _ \n _\n\n _\n\n _Robert _B oul ter _is _an _English _film _, _television _and _theatre
628 796 5199 347 2852 353 796 220 198 628 628 5199 347 2852 353 318 281 3594 2646 837 5581 290 21421
_actor _. . _He _had _a _guest _@ - @ _starring _role _on _the _television _series _The _Bill _in _2000 _. . _
8674 764 679 550 257 8319 2488 12 31 20495 2597 319 262 5581 2168 383 3941 287 4751 764 220

**0.0273**: \n\n _= _Robert _B oul ter _= _ _ \n\n _\n\n _\n _Robert _B oul ter _is _an _English _film _, _television _and _theatre
628 796 5199 347 2852 353 796 628 628 198 5199 347 2852 353 318 281 3594 2646 837 5581 290 21421
_actor _. . _He _had _a _guest _@ - @ _starring _role _on _the _television _series _The _Bill _in _2000 _. . _
8674 764 679 550 257 8319 2488 12 31 20495 2597 319 262 5581 2168 383 3941 287 4751 764 220

0.0178: \n\n _ _ = _Robert _B oul ter _= _ _ \n _\n _\n\n _\n _Robert _B oul ter _is _an _English _film _, _television _and _theatre
628 220 28 5199 347 2852 353 796 220 198 198 628 198 5199 347 2852 353 318 281 3594 2646 837 5581 290 21421
_actor _. . _He _had _a _guest _@ - @ _starring _role _on _the _television _series _The _Bill _in _2000 _. . _
8674 764 679 550 257 8319 2488 12 31 20495 2597 319 262 5581 2168 383 3941 287 4751 764 220

0.0061: \n\n _= _Robert _B oul ter _= _ _ \n _\n\n _\n _\n _Robert _B oul ter _is _an _English _film _, _television _and _theatre
628 796 5199 347 2852 353 796 220 198 628 198 198 5199 347 2852 353 318 281 3594 2646 837 5581 290 21421
_actor _. . _He _had _a _guest _@ - @ _starring _role _on _the _television _series _The _Bill _in _2000 _. . _
8674 764 679 550 257 8319 2488 12 31 20495 2597 319 262 5581 2168 383 3941 287 4751 764 220

0.0032: \n\n _= _Robert _B oul ter _= _ _ \n\n _\n _\n\n _Robert _B oul ter _is _an _English _film _, _television _and _theatre
628 796 5199 347 2852 353 796 220 628 198 628 5199 347 2852 353 318 281 3594 2646 837 5581 290 21421
_actor _. . _He _had _a _guest _@ - @ _starring _role _on _the _television _series _The _Bill _in _2000 _. . _
8674 764 679 550 257 8319 2488 12 31 20495 2597 319 262 5581 2168 383 3941 287 4751 764 220

0.0011: \n\n _= _Robert _B oul ter _= _ _ \n _\n _\n\n _\n _Robert _B oul ter _is _an _English _film _, _television _and _theatre
628 796 5199 347 2852 353 796 220 198 198 628 198 5199 347 2852 353 318 281 3594 2646 837 5581 290 21421
_actor _. . _He _had _a _guest _@ - @ _star ring _role _on _the _television _series _The _Bill _in _2000 _. . _
8674 764 679 550 257 8319 2488 12 31 3491 1806 2597 319 262 5581 2168 383 3941 287 4751 764 220

0.0002: \n\n _= _Robert _B oul ter _= _ _ \n _\n _\n\n _\n _Robert _B oul ter _is _an _English _film _, _television _and _theatre
628 796 5199 347 2852 353 796 220 198 198 628 198 5199 347 2852 353 318 281 3594 2646 837 5581 290 21421
_actor _. . _He _had _a _guest _ _ @ - @ _starring _role _on _the _television _series _The _Bill _in _2000 _ . _
8674 764 679 550 257 8319 220 31 12 31 20495 2597 319 262 5581 2168 383 3941 287 4751 220 13 220

**UTF-8 Encoding.**    The very first step in representing text is the choice of string encoding. UTF-8 is an encoding that maps strings of characters into bytes, which is used to store almost every webpage on the internet. The scheme uses a variable-width encoding of each Unicode character that uses between 1 and 4 bytes, with more common Unicode characters requiring fewer bytes. The UTF-8 encoding imposes additional validity constraints on token strings. In future work, we may extend our work on BPE encoding validity one step further to ensure that the byte string that the BPE string decodes is a valid UTF-8 string. For better or worse, UTF-8 decoding errors are typically suppressed, e.g., by the Huggingface transformers library.

**Implementation note.**    Byte-pair encoding (as the name suggests) typically takes *bytes* as inputs. Note that popular interfaces like huggingface try to make things easier for users by hiding the UTF-8 encoding step that maps strings to bytes

from users by exposing only a string interface. We require finer-grained control to implement the round-trip test correctly; we must avoid encoding and decoding from the text's encoding (e.g., UTF-8). Therefore, we use a custom implementation of byte-pair encoding. Note that the `transformers` library does not provide methods to work directly with byte strings.

`"🤗"`.encode(`"utf-8"`) = `b'\xf0\x9f\xa4\x97'` is the encoding of the hugging face emoji 🤗.

However, if we try to decode the following prefix of the byte string `b'\xf0\x9f\xa4'`.decode(`"utf-8"`), we get an error as it is an invalid UTF-8 string. Note that if we use `b'\xf0\x9f\xa4'`.decode(`"utf-8"`, errors=`"ignore"`) or `b'\xf0\x9f\xa4'`.decode(`"utf-8"`, `"replace"`) the errors may go unnoticed. Many packages (e.g., `transformers` and `tiktoken`) suppress these errors.

**Other tokenization models.**    We limited our experiments to BPE, as it is by far the most commonly used tokenization model (Footnote 5). However, many other tokenizers exist, such as longest-match tokenization (used by WordPiece) and other finite-state transduction-based tokenizers. These other tokenizers can plug directly into our framework as long as they provide efficient membership tests for $\overrightarrow{\mathcal{D}}$. In the case of finite-state-transduction methods (which include longest-match tokenizers), the membership is a straightforward operation on the automaton. We note that $\mathcal{D} \neq \overrightarrow{\mathcal{D}}$, in general—and, it specifically does not hold for longest-match tokenizers.

**More sophisticated inference algorithms.**    Future work may wish to explore *sequential* importance resampling (SIR) methods or other sequential Monte Carlo methods (SMC), which have shown great promise as methods for constrained generation from language models (e.g., Lew et al., 2023; Zhao et al., 2024). Sequential importance resampling, for example, provides the computational benefit of not waiting until a complete string is sampled from the local model to resample. It works by evolving many string prefixes in parallel. Now, when any given string is "stuck" in a low-probability region, it can be probabilistically replaced by another string in the collection. Many SMC algorithms, including SIR, are GPU-friendly.

**Distillation.**    Another interesting direction to explore related to fine-tuning is the distillation of the (approximately) globally canonicalized model $g$. We could do that by minimizing $\mathrm{KL}(g \| \ell_{\boldsymbol{\theta}})$ using stochastic gradient descent.

**Precision vs. recall.**    It can be challenging to correctly capture all of the constraints that the inputs must satisfy. In this paper, we explored the canonicality constraints that BPE models have. We provided efficient methods for testing canonicality, and we needed to validate those against reference canonicality procedures (as a kind of regression test). It may be impossible to capture every nuance of a particular implementation perfectly. And, in some cases, there are even "bugs" or "undesirable" behaviors that the reference implementations have. For example, the pre-tokenization system used by GPT-2 was designed to split the possessive marker ('s) from a phrase like `OpenAI's␣system` → `OpenAI|'s|␣system`; however, the rule has the following behavior which is likely not intensional `OpenAI␣Inc.'s␣system` → `OpenAI|␣Inc|.'|s|␣system`, as it fails to split the possessive correctly because the regular expression used by the pre-tokenizer prioritizes `.'` over `'s` in this example. More generally, reasoning about a complicated software system using formal models will always present a challenge between fidelity and analytical tractability.

**Evaluation on downstream tasks.**    Our study deliberately centers on a task-agnostic metric—KL divergence. We proved that our canonicalization methods are guaranteed to improve this metric (Propositions 1 and 3) and we quantified the effect empirically (§5). However, many language-model applications are judged with task-specific metrics that may be *misaligned* with KL divergence. Thus, an improvement in KL divergence does not guarantee an improvement in the task-specific metrics. Assessing how canonicalization influences concrete reasoning tasks—under each task's own evaluation metric—remains an important direction for future work.

# B. Efficient Membership Tests for BPE

This section gives a formal definition of byte-pair encoding (BPE) and novel algorithms for the following membership tests pertaining to it:

- the canonical set $\mathcal{D}$ (see Theorem 1)
- the canonical prefix set $\overrightarrow{\mathcal{D}}$ (see Lemma 3)
- an incremental membership test for single-token extensions, i.e., $\boldsymbol{\delta} \cdot \boldsymbol{\delta}' \in \overrightarrow{\mathcal{D}}$ given that $\boldsymbol{\delta} \in \overrightarrow{\mathcal{D}}$ (see Proposition 5)

Our key insight is that a token string is canonical if and only if all of its constituent bigrams are canonical; see Theorem 1. Thus, our membership test runs in time linear in the length of the string—we iterate over all bigrams in the strings and check their canonicality; we call this the **bigram test**. Moreover, our test is simple to implement, unlike similar tests given in prior work (e.g., Berglund et al., 2024; Cognetta & Okazaki, 2024). A consequence of Theorem 1 is that $\mathcal{D} = \overrightarrow{\mathcal{D}}$ (see Lemma 3), which means that membership in the set of canonical prefixes reduces to membership in the canonical set.[19]

## B.1. Byte-Pair Encoding

Byte-pair encoding (BPE; Gage, 1994; Sennrich et al., 2016) is currently the most commonly used tokenization method, as it has been used in most recent state-of-the-art language models.[20] We formalize BPE as follows.

**Definition 4.** *Byte-pair encoding is a pair* $(\Sigma, \mathcal{M})$ *that defines a tokenization model* $(\Sigma, \Delta, \tau, \kappa)$. *We describe the assumptions on* $(\Sigma, \mathcal{M})$ *and the construction of* $(\Sigma, \Delta, \tau, \kappa)$ *below:*

- $\Sigma$ *is a **base alphabet** of characters (or bytes)*
- $\mathcal{M} = \left[ \langle \boldsymbol{\sigma}'_{(1)}, \boldsymbol{\sigma}''_{(1)} \rangle, \dots, \langle \boldsymbol{\sigma}'_{(M)}, \boldsymbol{\sigma}''_{(M)} \rangle \right]$ *is the **merge list** where* $\boldsymbol{\sigma}'_{(m)}, \boldsymbol{\sigma}''_{(m)} \in \Sigma^+$ *for* $m = 1, \dots, M$.
- *We define the **subword alphabet** $\underline{S} \overset{\text{def}}{=} \Sigma \cup \{\boldsymbol{\sigma}' \cdot \boldsymbol{\sigma}'' \mid \langle \boldsymbol{\sigma}', \boldsymbol{\sigma}'' \rangle \in \mathcal{M}\}$.*
- *We define the **token alphabet** $\Delta$ such that for each $\delta \in \Delta$, there is a unique object (e.g., an identifier) for each distinct element of the subword alphabet $\underline{S}$. The reason for this technical detail is to avoid confusion between the concatenation of subwords and the concatenation of tokens.*
- *We define the **base encoder** $\underline{\tau} \colon \underline{S} \to \underline{\Delta}$ and **base decoder** $\underline{\kappa} \colon \underline{\Delta} \to \underline{S}$ as bijective maps between the subword alphabet $\underline{S}$ and a set of **base tokens** $\underline{\Delta} \subseteq \Delta$.*
- *We define the **decoder** $\kappa \colon \Delta^* \to \Sigma^*$ to be the pointwise extension of the base decoder: $\kappa(\delta_1 \cdots \delta_N) \overset{\text{def}}{=} \underline{\kappa}(\delta_1) \cdots \underline{\kappa}(\delta_N)$.*
- *We define the **encoder** $\tau \colon \Sigma^* \to \Delta^*$ in the pseudocode, which is a deterministic string rewriting process based on applying a sequence of merge rules to the current string of tokens until no more merges are available.[21],[22]*

```
14  def τ(σ₁ ⋯ σ_N):
15      δ ← τ(σ₁) ⋯ τ(σ_N)
16      while True:
17          δ' ← rewrite(δ)
18          if δ' = δ: return δ
19          δ ← δ'
```

```
20  def rewrite(δ₁ ⋯ δ_N):
21      for ⟨σ', σ''⟩ in M
22          for n in [1, ... , N−1]:
23              if ⟨κ(δ_n), κ(δ_{n+1})⟩ = ⟨σ', σ''⟩:
24                  return δ₁ ⋯ δ_{n−1} τ(σ' · σ'') δ_{n+2} ⋯ δ_N
25      return δ₁ ⋯ δ_N
```

*Note that the merge rules are applied in a specific deterministic order to any given string; thus, there is a unique way to derive the output tokenization from the input string.*

---

[19]We note that, in principle, the bigram canonicality tests can all be precomputed and cached for maximal efficiency. However, we found that to be impractical for models with large token vocabularies. In App. B.3, we provide a very efficient method that can test bigram canonicality.

[20]See Footnote 5.

[21]We note that $\tau$ can be implemented much more efficiently using a careful choice of data structure (see, e.g., Zouhar et al., 2023).

[22]We note that Berglund et al. (2024) refers to this incarnation of BPE as the *SentencePiece* BPE algorithm.

## B.2. Simple and Efficient Membership Tests

**Definition 5.** *We define the **set of canonical bigrams** as*

$$\mathcal{B} \stackrel{\text{def}}{=} \{\delta \cdot \delta' \mid \delta, \delta' \in \Delta, \varphi(\delta \cdot \delta') = \delta \cdot \delta'\}. \tag{19}$$

*where $\varphi$ is the canonicalization function (§2.6).*

**Definition 6.** *We define the **set of bigram-canonical strings** $B$ as strings composed of only canonical bigrams:*

$$B \stackrel{\text{def}}{=} \{\boldsymbol{\delta} \in \Delta^* : \text{BIGRAMS}(\boldsymbol{\delta}) \subseteq \mathcal{B}\}. \tag{20}$$

In general, it is possible for a token in the token vocabulary not to be canonical. Such tokens can always be removed from the token vocabulary without changing the result of the BPE tokenizer. We assume, for simplicity, that the token vocabulary contains no such tokens.

**Assumption 1.** *For all $\delta \in \Delta$, $\varphi(\delta) = \delta$.*

**Definition 7.** *We define the **truncated canonicalization function** $\varphi_T$ as follows:*

```
26  def φ_T(δ):
27    σ_1 ··· σ_N ← κ(δ)              # decode token string
28    δ^(0) ← τ(σ_1) ··· τ(σ_N)       # initialize with base tokens
29    for t = 0, ... , (T − 1):
30      δ^(t+1) ← rewrite(δ^(t))
31      if δ^(t+1) = δ^(t): return δ^(t)
32    return δ^(T)
```

**Remark 1.** *It is straightforward to verify that $\varphi = \varphi_\infty$, as it is equivalent to code for $\tau(\kappa(\cdot))$ where we have added the option of truncating the number of iterations performed in $\tau$'s fixpoint iteration loop. Setting $T = \infty$, disables the truncation; thus, $\varphi = \varphi_\infty$.*

**Lemma 1** (Canonicalization process synchronization). *Suppose that $abc \in \mathcal{D}$ or both $ab, bc \in \mathcal{D}$. Suppose further that $b$ is nonempty. Then, there exists a sequence $\{(a_t, b_t, c_t, d_t, e_t)\}_{t=1}^{\infty}$ such that the following equations hold for all $t \geq 0$:*

$$P_t \stackrel{\text{def}}{=} \begin{cases} \varphi_t(abc) = \varphi_{a_t}(a)\varphi_{b_t}(b)\varphi_{c_t}(c) \\ \varphi_{d_t}(ab) = \varphi_{a_t}(a)\varphi_{b_t}(b) \\ \varphi_{e_t}(bc) = \varphi_{b_t}(b)\varphi_{c_t}(c) \end{cases} \tag{21}$$

*Proof.* We will prove Lemma 1 by induction on $t$, and by explicitly constructing the sequence $\{(a_t, b_t, c_t, d_t, e_t)\}_{t=1}^{\infty}$.

Let $T$ denote the number of distinct iterations of $\varphi(abc)$.

*Base case ($P_0$).* In this case, it is straightforward to verify that $(a_0, b_0, c_0, d_0, e_0) = (0, 0, 0, 0, 0)$ satisfies $P_0$ because each canonicalization call returns precisely the base tokenization of its argument. Thus, $P_0$ holds.

*Induction step.* Suppose $P_0, ..., P_j$ hold. We seek to show that $P_{j+1}$ holds.

Since $P_j$ holds, there must exist $(a_t, b_t, c_t, d_t, e_t)$ satisfying the equations in $P_t$. To prove that $P_{t+1}$ holds, we will construct $(a_{t+1}, b_{t+1}, c_{t+1}, d_{t+1}, e_{t+1})$ such that the equations in $P_{t+1}$.

We first observe that at time $t$, $\varphi_t(abc) = a^{(t)}b^{(t)}c^{(t)}$ where

- $a^{(t)}$ is the tokenization at step $t$ of the substring over the characters in $a$ in the canonicalization process for $abc$
- $b^{(t)}$ is the tokenization at step $t$ of the substring over the characters in $b$ in the canonicalization process for $abc$
- $c^{(t)}$ is the tokenization at step $t$ of the substring over the characters in $c$ in the canonicalization process for $abc$

The only way for this observation to be false is for a merge to straddle the boundary between $a^{(t)}$ and $b^{(t)}$ or $b^{(t)}$ and $c^{(t)}$, which is impossible because of our premise that $abc \in \mathcal{D}$ or both $ab, bc \in \mathcal{D}$. More specifically, if there ever were a merge between $a^{(t)}$ and $b^{(t)}$ then $ab$ and $abc$ could not be canonical, as the final tokenization would not respect the boundary between $a$ and $b$. Similarly, if there ever were a merge between $b^{(t)}$ and $c^{(t)}$, then by analogous reasoning, $bc$ and $abc$ could not be canonical.

We now turn to case analysis:

1. Suppose $t < T$, then the $(t+1)^{\text{th}}$ step of $\varphi(abc)$ applies to the highest-priority[23] bigram of $a^{(t)}b^{(t)}c^{(t)}$. Consider the following subcases characterizing the possible positions for this merge:
   (a) The merge is in $a^{(t)}$.
      Then, $(a_{t+1}, b_{t+1}, c_{t+1}, d_{t+1}, e_{t+1}) = (a_t + 1, b_t, c_t, d_t + 1, e_t)$ satisfies $P_{t+1}$ because
      - It must also be the highest-priority merge in $a^{(t)}$, which is step $a_{t+1}$ of $\varphi(a)$.
      - It must also be the highest-priority merge in $a^{(t)}b^{(t)}$, which is step $d_{t+1}$ of $\varphi(ab)$.
      - The other canonicalization processes are unaffected, so they copy their position on this step.
   (b) The merge is in $b^{(t)}$.
      Then, $(a_{t+1}, b_{t+1}, c_{t+1}, d_{t+1}, e_{t+1}) = (a_t, b_t + 1, c_t, d_t + 1, e_t + 1)$ satisfies $P_{t+1}$ because
      - It must also be the highest-priority merge in $b^{(t)}$, which is step $b_{t+1}$ of $\varphi(b)$.
      - It must also be the highest-priority merge in $a^{(t)}b^{(t)}$, which is step $d_{t+1}$ of $\varphi(ab)$.
      - It must also be the highest-priority merge in $b^{(t)}c^{(t)}$, which is step $e_{t+1}$ of $\varphi(bc)$.
      - The other canonicalization processes are unaffected, so they copy their position on this step.
   (c) The merge is in $c^{(t)}$.
      Then, $(a_{t+1}, b_{t+1}, c_{t+1}, d_{t+1}, e_{t+1}) = (a_t, b_t, c_t + 1, d_t, e_t + 1)$ satisfies $P_{t+1}$ because
      - It must also be the highest-priority merge in $c^{(t)}$, which is step $c_{t+1}$ of $\varphi(c)$.
      - It must also be the highest-priority merge in $b^{(t)}c^{(t)}$, which is step $e_{t+1}$ of $\varphi(bc)$.
      - The other canonicalization processes are unaffected, so they copy their position on this step.
2. Suppose $t \geq T$. No merges exist.
   Then, $(a_{t+1}, b_{t+1}, c_{t+1}, d_{t+1}, e_{t+1}) = (a_t, b_t, c_t, d_t, e_t)$ satisfies $P_{t+1}$ because $\varphi_t(abc) = \varphi_T(abc) = a^{(t)}b^{(t)}c^{(t)} = a^{(T)}b^{(T)}c^{(T)}$, by definition of $T$, which implies that no further changes are possible and all processes must copy their position on this step, and continue to do so forever.

Therefore, $P_t$ holds for all $t \geq 0$ by the principle of induction. ∎

---

[23]I.e., the bigram with the lexicographically lowest pair of (position in the merge list, position in the current token string). Note that there can be no ties under this ordering.

**Example 1.** *This example seeks to illustrate Lemma 1. Consider the following three token strings from the GPT-2 tokenizer:*

$$a = \underset{260}{\text{re}} \cdot \underset{12}{\text{-}} \qquad b = \underset{30001}{\text{token}} \qquad c = \underset{1143}{\text{ized}}$$

*For these tokens, we have $\kappa(abc) = \text{re-tokenized}$. Our worked example will focus on only the first equation in $P_t$ (repeated below for convenience), as the other two equations behave very similarly:*

$$\varphi_t(abc) = \varphi_{a_t}(a)\varphi_{b_t}(b)\varphi_{c_t}(c)$$

*Before moving forward with the example, we note that each token during the canonicalization process will be rendered as a tree because it is a convenient representation for showing which merges were done to build each of them.[24] For example, our inputs $a$, $b$, and $c$ are represented by the following trees:*

*Each row in the table below corresponds to a time step $t$, and the content of each column is the status of the canonicalization process at that time. We mark cells that are copied from the cell immediately above them with a dash (—).*

We see that at every step $t$, the equation $\varphi_t(abc) = \varphi_{a_t}(a)\varphi_{b_t}(b)\varphi_{c_t}(c)$ is satisfied for an appropriate choice of $(a_t, b_t, c_t)$. Specifically, we count the number of non-copy actions in each row. In the example, for $t = 8$, we have $(a_t, b_t, c_t) = (1, 4, 3)$. This example also allows us to see the case analysis used in the proof for the *location of the merge* in action.

---

[24]We give a more formal presentation of these trees in App. B.3.

**Lemma 2** (Interlocking Canonicalization).

$$\forall a \in \Delta^*, b \in \Delta^+, c \in \Delta^*: abc \in \mathcal{D} \iff ab \in \mathcal{D} \land bc \in \mathcal{D} \tag{22}$$

*Proof.* Fix $a \in \Delta^*$, $b \in \Delta^+$, and $c \in \Delta^*$ arbitrarily.

We will consider the direction of the bi-implication separately.

**Case** ($\implies$). Suppose that $abc \in \mathcal{D}$. Then, Lemma 1 gives us a sequence $\{(a_t, b_t, c_t, d_t, e_t)\}_{t=1}^{\infty}$ such that the following equations hold for all $t \geq 0$.

$$P_t \overset{\text{def}}{=} \begin{cases} \varphi_t(abc) = \varphi_{a_l}(a)\varphi_{b_t}(b)\varphi_{c_t}(c) \\ \varphi_{d_t}(ab) = \varphi_{a_t}(a)\varphi_{b_t}(b) \\ \varphi_{e_t}(bc) = \varphi_{b_t}(b)\varphi_{c_t}(c) \end{cases} \tag{23a}$$

Now, consider $(a_T, b_T, c_t, d_T, e_T)$, which gives

$$\begin{cases} \varphi_T(abc) = \varphi_{a_T}(a)\varphi_{b_T}(b)\varphi_{c_T}(c) \\ \varphi_{d_T}(ab) = \varphi_{a_T}(a)\varphi_{b_T}(b) \\ \varphi_{e_T}(bc) = \varphi_{b_T}(b)\varphi_{c_T}(c) \end{cases} \tag{23b}$$

which implies the following because, at time $T$, *each* of the canonicalization processes has reached its respective fixpoint:

$$\begin{cases} \varphi(abc) = \varphi(a)\varphi(b)\varphi(c) \\ \varphi(ab) = \varphi(a)\varphi(b) \\ \varphi(bc) = \varphi(b)\varphi(c) \end{cases} \tag{23c}$$

which implies the following because each call to $\varphi$ must return the canonicalization of its respective argument:

$$\begin{cases} \varphi(abc) = abc \\ \varphi(ab) = ab \\ \varphi(bc) = bc \end{cases} \tag{23d}$$

Therefore, $ab$ and $bc$ are also canonical.

**Case** ($\impliedby$). Suppose $ab, bc \in \mathcal{D}$. Then, following an identical argument as the $\implies$-case, we can see that $abc \in \mathcal{D}$.

Thus, the bi-implication holds; thus, Lemma 2 holds. ■

Theorem 1 establishes that a BPE token string is canonical if and only if all of its bigrams are canonical.[25]

**Theorem 1.** $\mathcal{D} = B$.

*Proof.* We seek to prove, for all $\boldsymbol{\delta} \in \Delta^*$, that $\boldsymbol{\delta} \in \mathcal{D} \iff \boldsymbol{\delta} \in B$. We do this using induction on prefixes of $\boldsymbol{\delta}$ and Lemma 2.

Choose $\boldsymbol{\delta} = \delta_1 \cdots \delta_N \in \Delta^* \in \Delta^*$ arbitrarily. Let the induction hypothesis $P(i)$ be that $\boldsymbol{\delta}_{\leq i} \in \mathcal{D} \iff \boldsymbol{\delta}_{\leq i} \in B$.

*Base cases.* We first prove $P(0)$ and, if $|\boldsymbol{\delta}| \geq 1$, $P(1)$.

- **Case $P(0)$.** To prove $P(0)$, we seek to show that $\boldsymbol{\delta}_{\leq 0} = \varepsilon \in \mathcal{D} \iff \boldsymbol{\delta}_{\leq 0} = \varepsilon \in B$. Both sides of the bi-implication are always true: $\varepsilon \in \mathcal{D}$ because $\varphi(\varepsilon) = \tau(\kappa(\varepsilon)) = \tau(\varepsilon) = \varepsilon$; and $\varepsilon \in B$ because it contains no bigrams, hence it is vacuously true. Therefore, $P(0)$ is true.
- **Case $P(1)$.** To prove that $P(1)$ is true for cases where $|\boldsymbol{\delta}| \geq 1$, we seek to show that $\boldsymbol{\delta}_{\leq 1} = \delta_1 \in \mathcal{D} \iff \boldsymbol{\delta}_{\leq 1} = \delta_1 \in B$. Again, both sides of the bi-implication are always true. By Assumption 1, we know that $\delta_1 \in \mathcal{D}$ is true. Since $\delta_1$ has no bigrams, $\delta_1 \in B$ is vacuously true. Therefore, $P(1)$ is true.

*Induction step.* Assume $P(i)$, that $\boldsymbol{\delta}_{\leq i} \in \mathcal{D} \iff \boldsymbol{\delta}_{\leq i} \in B$, where $i \geq 2$. We seek to prove $P(i+1)$, that $\boldsymbol{\delta}_{\leq i+1} \in \mathcal{D} \iff \boldsymbol{\delta}_{\leq i+1} \in B$. We prove each direction of the bi-implication separately.

- **Forward.** We seek to prove $\boldsymbol{\delta}_{\leq i+1} \in \mathcal{D} \implies \boldsymbol{\delta}_{\leq i+1} \in B$. Assume $\boldsymbol{\delta}_{\leq i+1} = \boldsymbol{\delta}_{\leq i-1}\delta_i\delta_{i+1} \in \mathcal{D}$. By Lemma 2, we know that $\boldsymbol{\delta}_{\leq i-1}\delta_i \in \mathcal{D}$ and $\delta_i\delta_{i+1} \in \mathcal{D}$. Since $\boldsymbol{\delta}_{\leq i-1}\delta_i \in \mathcal{D}$, by $P(i)$, we know that $\boldsymbol{\delta}_{\leq i-1}\delta_i \in B$. Since $\boldsymbol{\delta}_{\leq i-1}\delta_i \in B$ and $\delta_i\delta_{i+1} \in \mathcal{D}$, all of the bigrams in $\boldsymbol{\delta}_{\leq i-1}\delta_i\delta_{i+1} = \boldsymbol{\delta}_{\leq i+1}$ are canonical, so $\boldsymbol{\delta}_{\leq i+1} \in B$.
- **Backward.** We seek to prove $\boldsymbol{\delta}_{\leq i+1} \in B \implies \boldsymbol{\delta}_{\leq i+1} \in \mathcal{D}$. Assume $\boldsymbol{\delta}_{\leq i+1} = \boldsymbol{\delta}_{\leq i-1}\delta_i\delta_{i+1} \in B$. Then $\boldsymbol{\delta}_{\leq i-1}\delta_i \in B$, because it contains only a subset of the bigrams in $\boldsymbol{\delta}_{\leq i-1}\delta_i\delta_{i+1}$. Then, by $P(i)$, we know that $\boldsymbol{\delta}_{\leq i-1}\delta_i \in \mathcal{D}$. From $\boldsymbol{\delta}_{\leq i-1}\delta_i\delta_{i+1} \in B$ and Def. 6, we know that $\delta_i\delta_{i+1} \in \mathcal{D}$. Since $\boldsymbol{\delta}_{\leq i-1}\delta_i \in \mathcal{D}$ and $\delta_i\delta_{i+1} \in \mathcal{D}$, by Lemma 2, we know that $\boldsymbol{\delta}_{\leq i-1}\delta_i\delta_{i+1} = \boldsymbol{\delta}_{\leq i+1} \in \mathcal{D}$.

By induction, we know $P(N)$ is true, so $\boldsymbol{\delta} \in \mathcal{D} \iff \boldsymbol{\delta} \in B$, and the proof is complete. ∎

---

[25]Note that if a string with length less than or equal to one has no bigrams; thus, the property is trivially satisfied, i.e., all length-zero and length-one token strings are canonical. The latter is thanks to Assumption 1.

**Lemma 3.** *For BPE,[26] the set of canonical strings is prefix closed, i.e., $\mathcal{D} = \overrightarrow{\mathcal{D}}$.*

*Proof.* We prove $\mathcal{D} \subseteq \overrightarrow{\mathcal{D}}$ and $\overrightarrow{\mathcal{D}} \subseteq \mathcal{D}$.

- *Part 1* ($\mathcal{D} \subseteq \overrightarrow{\mathcal{D}}$). Obvious by definition of $\overrightarrow{\mathcal{D}}$.
- *Part 2* ($\overrightarrow{\mathcal{D}} \subseteq \mathcal{D}$).
  1. Suppose $\delta \in \overrightarrow{\mathcal{D}}$
  2. $\implies$ by definition of $\overrightarrow{\mathcal{D}}$, there is a $\delta' \in \mathcal{D}$ such that $\delta \preceq \delta'$
  3. $\implies$ by Theorem 1, the bigrams of $\delta'$ are all canonical
  4. $\implies$ since $\delta \preceq \delta'$, its bigrams are a subset of those of $\delta'$, so they are also all canonical
  5. $\implies$ since the bigrams of $\delta$ are all canonical, by Theorem 1, $\delta \in \mathcal{D}$

Therefore, $\overrightarrow{\mathcal{D}} = \mathcal{D}$. ∎

---

[26]More generally, Lemma 3 holds for any tokenization model with a bigram-based canonicality test (i.e., $\delta \in \mathcal{D} \iff \text{BIGRAMS}(\delta) \subseteq \mathcal{B}$.

## B.3. An *Even Faster* Bigram Test

**BPE derivations.** Our canonicality test involves the inspection of the merge structure within a tokenization of a string. For that purpose, we define the **derivation** of a given string $\gamma(\boldsymbol{\sigma})$ as a string of trees $t_1 \cdots t_N$ where each $t_n$ (for $n = 1, \dots, N$) is a binary tree. For our purposes, a (binary) **tree** is a recursive data type composed of either a pair of trees or a base token. The pseudocode below shows how $\tau$ can be augmented to produce these derivation trees for a given string:

```
33 def γ(σ₁ ⋯ σ_N):
34     t ← τ(σ₁) ⋯ τ(σ_N)
35     while True:
36         t′ ← step_derivation(t)
37         if t′ = t: return t
38         t ← t′
```

```
39 def step_derivation(t₁ ⋯ t_N):
40     for ⟨σ′, σ″⟩ ∈ M:
41         for n ∈ [1, … N − 1]:
42             if ⟨κ(t_n), κ(t_{n+1})⟩ = ⟨σ′, σ″⟩:
43                 return t₁ ⋯ t_{n−1}⟨t_n, t_{n+1}⟩ t_{n+2} ⋯ t_N
44     return t₁ ⋯ t_N
```

Note that $\gamma(\cdot)$ is an adaptation of $\tau$. The difference is that $\underline{\tau}$ has been replaced with a function that creates trees instead of replacing them with tokens. We have also extended $\underline{\kappa}$ to return the subword denoted by the derivation.

For each token $\delta \in \Delta$, we associate with it a unique **canonical derivation** $\gamma(\delta)$. However, we note that when $|\gamma(\delta)| > 1$, it means that the token $\delta$ does not canonically tokenize to itself. Assumption 1 ensures $|\gamma(\delta)| = 1$ for all $\delta \in \Delta$.

**Additional notation.** For notational convenience, we define the following:

- For any $\ell \in \Sigma \cup \Sigma^+ \times \Sigma^+$, let $\mathcal{M}[\ell]$ denote the following:
  - If $\ell \in \Sigma$, return its position in some fixed, arbitrary ordering of $\Sigma$.
  - If $\ell \in \Sigma^+ \times \Sigma^+$, return the rank position of the merge $\ell$ in the merge list plus $|\Sigma|$ (as an offset), and $\infty$ if $\ell$ is not in $\mathcal{M}$.
- We extend $\mathcal{M}[t]$ to trees $t$ as follows.
  - For a tree in $t \in \underline{\Delta}$: $\mathcal{M}[t] \stackrel{\text{def}}{=} \mathcal{M}[\underline{\kappa}(t)]$.
  - For a tree of the form $t = \langle s, s' \rangle$: $\mathcal{M}[t] \stackrel{\text{def}}{=} \mathcal{M}[\langle \underline{\kappa}(s), \underline{\kappa}(s') \rangle]$.
- We define the following **ordering relations** on merge rules and subtrees: We define $\ell < r \iff \mathcal{M}[\ell] < \mathcal{M}[r]$. We define $\ell > r$, $\ell \leq r$, and $\ell \geq r$ analogously.
- Given a token $\delta \in \Delta$, we define the **left spine** $\mathcal{S}(\gamma(\delta))$ of $\gamma(\delta)$ to be the ordered set containing the root node and its left descendants[27] (ordered from root to leaf); and, we define the **right spine** $\mathcal{R}(\gamma(\delta))$ analogously (for right descendants).
- In a given tree, we define $\pi$ as the function that returns the parent node for any of its subtrees.

**Intuition.** At a high level, we say that a token $\delta''$ *conflicts* with a bigram $\delta \cdot \delta'$ if and only if $\delta''$ applies with an earlier rank over any overlapping span of the same character string. More formally, we say that $\delta''$ (nontrivially) **overlaps** with $\delta$ and $\delta'$ if and only if

$$\kappa(\delta) = \sigma_1 \cdots \sigma_n \tag{24a}$$

$$\kappa(\delta') = \sigma_{n+1} \cdots \sigma_N \tag{24b}$$

$$\kappa(\delta \cdot \delta') = \sigma_1 \cdots \sigma_n \cdot \sigma_{n+1} \cdots \sigma_N \tag{24c}$$

$$\kappa(\delta'') = \sigma_i \cdots \sigma_j \quad \text{where } 1 \leq i \leq n < j \leq N \tag{24d}$$

However, for $\delta''$ to conflict with $\delta \cdot \delta'$, it would need to include a merge over the span $\sigma_i \cdots \sigma_j$ that blocks at least one merge in $\delta$ and $\delta'$ from occurring in $\varphi(\delta \cdot \delta')$.

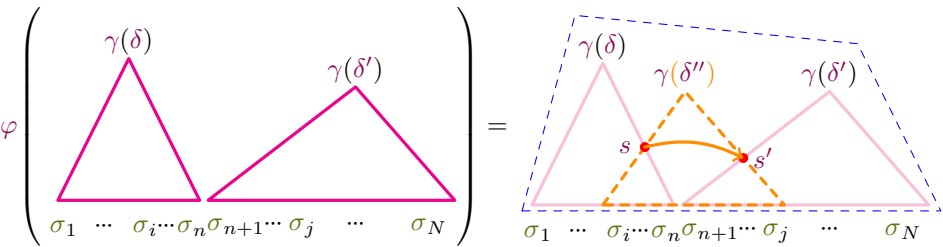

---

[27]We say that a node $p$ is a *left descendant* of a node $q$ if $p$ is the left child of $q$ or a left descendant of the left child of $q$. We define the right descendants of $p$ in an analogous manner.

In the diagram above, the derivation for the bigram $\delta \cdot \delta'$ is denoted by the juxtaposition of two triangles, one for each token's derivation tree. We have denoted the canonical derivation as a trapezoid, as it is generally the union of multiple triangles; there may be 1, 2, or more tokens in the canonicalized bigram $\varphi(\delta \cdot \delta')$.[28] In orange, we have drawn a third triangle for the conflicting token $\delta''$. We note that $\gamma(\delta'')$ must intersect $\gamma(\delta)$ at a node $s$ along its right-spine $\mathcal{R}(\gamma(\delta))$, and it must intersect $\gamma(\delta')$ at a node $s'$ along its left-spine $\mathcal{S}(\gamma(\delta'))$. In the diagram, the $s$ and $s'$ represent the subtrees at respective points of the intersection with the hypothetical conflicting tree. The orange arc from $s$ to $s'$ is used to show that any two nodes along the left and right edges could, in principle, be a conflicting tree. Crucially, however, the rank of the merge pair must precede the competing merge pairs present in the existing tokens $\delta$ and $\delta'$. We make this more precise below.

**Definition 8.** *A token $\delta''$ **conflicts** with the bigram $\delta \cdot \delta'$ if and only if $\gamma(\delta'')$ is a subtree of $\gamma(\varphi(\delta \cdot \delta'))$ (which may be a string of trees) but not a subtree of either $\gamma(\delta)$ or $\gamma(\delta')$.*

We can find a conflicting token by exploiting the structure of the tokenization algorithm.

**Definition 9.** *Let $\delta''$ be a token such that $\gamma(\delta'') = \langle s, s' \rangle$ for some $s \in \mathcal{R}(\gamma(\delta))$ and $s' \in \mathcal{S}(\gamma(\delta'))$. We say that $\delta''$ is a **minimal conflict** for the bigram $\delta \cdot \delta'$ if and only if $\pi(s) > \langle s, s' \rangle \leq \pi(s')$.*

The reason why $\delta''$ identified by Def. 9 is *minimal* is that its definition identifies the earliest possible deviation, i.e., the earliest merge rule that could apply on their shared string $\sigma_1 \cdots \sigma_N$ that was not in the existing merges in $\delta$ and $\delta'$. Note that the asymmetry $<$ versus $\leq$ is precisely what breaks ties between equal rank merges in favor of the left-most match, as prescribed in the procedure for $\tau$ that was provided in Def. 4.

Below is an algorithm for finding the minimal conflicting tree between $\delta$ and $\delta'$, if one exists.

```
45  def find_conflict(δ, δ′):
46      t ← γ(δ)
47      L ← ∞
48      while True:
49          t′ ← γ(δ′)
50          R ← ∞
51          while True:
52              if L > 𝓜[⟨t,t′⟩] ≤ R: return ⟨t,t′⟩    # conflict
53              if κ(t′) ∈ Δ: break   # t′ is a leaf
54              R ← 𝓜[t′]
55              ⟨t′,_⟩ ← t′    # descend the left spine
56          if t ∈ Δ: break   # t is a leaf
57          L ← 𝓜[t]
58          ⟨_,t⟩ ← t    # descend the right spine
59      return None
```

The `find_conflict` algorithm finds the minimal conflict between a pair of tokens $\delta$ and $\delta'$ (if one exists). We note that a useful consequence of Assumption 1 is that we do not need to check subtrees that exist off of the spines for conflicts, making our procedure more efficient than computing the canonicalized string from scratch, e.g., by running $\tau$ on the decoded string $\kappa(\delta \cdot \delta')$. Instead, our algorithm searches for a minimal conflicting tree: either it stops once it has found such a tree, or it has exhausted the limited options for such a tree. Finding the minimal conflict (if it exists) is *also* faster than computing the entire canonical tokenization $\varphi(\delta \cdot \delta')$, as it only requires finding the first difference rather than completely building the canonicalized bigram.[29]

**Definition 10.** *We define $\Phi \subseteq \Delta \times \Delta$ as the following relation on pairs of tokens:*

$$\Phi(\delta, \delta') \stackrel{\text{def}}{=} (\texttt{find\_conflict}(\delta, \delta') = \texttt{None}) \tag{25}$$

---

[28]Please refer back to Fig. 2 for examples.

[29]We note that the `find_conflict` algorithm can be efficiently vectorized so that it computes the complete vector of valid next tokens very efficiently. We will provide a PyTorch implementation in our public code release (upon publication).

The proposition below establishes that $\Phi$ provides a correct membership test for the set of canonical bigrams $\mathcal{B}$.

**Proposition 5.** $\Phi(\delta, \delta') \iff \delta \cdot \delta' \in \mathcal{B}$

*Proof.* Observe that $(\Phi(\delta, \delta') \iff \delta \cdot \delta' \in \mathcal{B}) \equiv (\neg\Phi(\delta, \delta') \implies \delta \cdot \delta' \notin \mathcal{B}) \land (\neg\Phi(\delta, \delta') \impliedby \delta \cdot \delta' \notin \mathcal{B})$. We will prove the proposition by proving each conjunct separately.

**Part 1** ($\neg\Phi(\delta, \delta') \implies \delta \cdot \delta' \notin \mathcal{B}$)**.**

1. Suppose $\neg\Phi(\delta, \delta')$. Then, there exists a conflicting pair $\langle s, s' \rangle = \texttt{find\_conflict}(\delta, \delta')$, i.e., $\langle s, s' \rangle$ satisfies $\pi(s) > \langle s, s' \rangle \leq \pi(s')$. Below is a schematic representation of such a conflict:

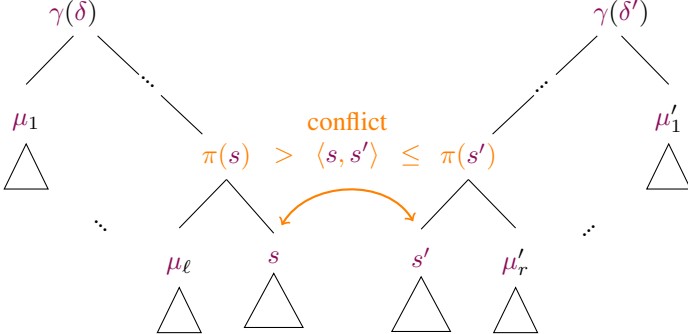

2. Thus, the conflicting merge $\langle s, s' \rangle$ would have been preferred by $\tau$ if it were run on the character string $\kappa(\delta \cdot \delta')$:

$$\langle s, s' \rangle$$
$$\mu_1 \quad \mu_\ell \quad \bigwedge \quad \mu_r' \quad \mu_1'$$
$$\triangle \cdots \triangle \; s \; s' \; \triangle \cdots \triangle$$

the conflicting merge *blocks* both of the merges below because $\pi(s) = \langle \mu_\ell, s \rangle > \langle s, s' \rangle$ and $\pi(s') = \langle s', \mu_r' \rangle \geq \langle s, s' \rangle$:[30]

$$\pi(s) \quad \pi(s')$$
$$\mu_1 \quad \bigwedge \quad \bigwedge \quad \mu_1'$$
$$\triangle \cdots \mu_\ell \; s \quad s' \; \mu_r' \cdots \triangle$$

3. The existence of this intermediate step means that it is impossible for $\delta \cdot \delta'$ to be a canonical bigram; thus, $\delta \cdot \delta' \notin \mathcal{B}$.

**Part 2** ($\neg\Phi(\delta, \delta') \impliedby \delta \cdot \delta' \notin \mathcal{B}$)**.**

1. Suppose $\delta \cdot \delta' \notin \mathcal{B}$. Then, $\varphi(\delta \cdot \delta') \neq \delta \cdot \delta'$, by definition.
2. Then, there must exist a conflicting merge $\langle s, s' \rangle$ as a subtree in $\varphi(\delta \cdot \delta')$ that blocks $\delta \cdot \delta'$ from being built. Now, because both $\delta$ and $\delta'$ are canonical in isolation (Assumption 1), the conflict $\langle s, s' \rangle$ must straddle the boundary between the $\delta$ and $\delta'$:

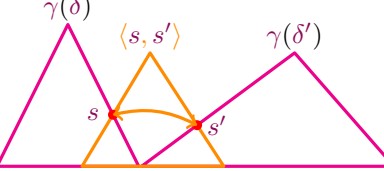

This implies that $s$ must be a subtree along the right edge of $\delta$ and $s'$ must be a subtree along the left edge of $\delta'$, as those are precisely the only ways that the straddling merge can occur under our circumstances. Lastly, for $\langle s, s' \rangle$ to block $\delta$ and $\delta'$, it would need to satisfy $\pi(s) > \langle s, s' \rangle \leq \pi(s')$.
3. This characterization is equivalent to $\neg\Phi(\delta, \delta')$.

$\blacksquare$

---

[30]Note that $\langle s, s' \rangle < \pi(s)$ enforces the left-most merge preference in BPE's encoding procedure $\tau$, i.e., in the event of a tie between overlapping merges of the same rank, the left-most merge is taken.

## B.4. Connections to Canonicality Algorithms

**Definition 11.** *We can describe the set of canonical tokenizations with the following finite-state automaton:*

- *$states = \Delta$*
- *$alphabet = \Delta$*
- *$transitions = \{\delta \xrightarrow{\delta'} \delta' \mid \delta \cdot \delta' \in \mathcal{B}\}$*
- *$initial = \Delta$*
- *$accepting = \Delta$*

Note that we never construct the automaton.

**Generality.** Our method works under a broader set of assumptions than Berglund et al. (2024); Lindmark (2024), as they require the merge list to be *proper*. This condition is, in fact, restrictive: the Llama tokenizer (used in our experiments) is not proper, but the GPT-2 tokenizer is proper.

**Practicality.** The automaton has $V$ nodes and $\leq V^2$ edges where $V = |\Delta|$. In practice, the machine is very close to having $V^2$ edges (even after DFA minimization!) because the transitions do not rule out many tokens. This means that the explicit machine is enormous. For example, GPT2 has 50K states and nearly 2,500,000,000 edges, making it way too large to build. Storing the edges that are *absent is significantly more efficient*. On the other hand, our version never builds the machine, so the memory complexity is not a problem. It is also possible to use a more sophisticated automata representation that allows for *failure transitions*; we refer the reader to Cognetta & Okazaki (2024); Lindmark (2024) for further discussion of the technique.

## C. Supporting Proofs and Lemmata

**Proposition 6.** *Suppose that the tokenization model $(\tau, \kappa, \Sigma, \Delta)$ is exact, then*

$$\forall \boldsymbol{\delta} \in \Delta^* : \ \boldsymbol{\delta} \in \mathcal{D} \iff \varphi(\boldsymbol{\delta}) = \boldsymbol{\delta} \tag{26}$$

*Proof.* Fix an arbitrary $\boldsymbol{\delta} \in \Delta^*$. We consider each direction separately:

- **Part 1** ($\boldsymbol{\delta} \in \mathcal{D} \implies \varphi(\boldsymbol{\delta}) = \boldsymbol{\delta}$):
  1. Suppose $\boldsymbol{\delta} \in \mathcal{D}$
  2. $\implies \exists \boldsymbol{\sigma} \in \Sigma^* : \boldsymbol{\delta} = \tau(\boldsymbol{\sigma})$ by definition of $\mathcal{D}$
  3. $\implies \kappa(\tau(\boldsymbol{\sigma})) = \boldsymbol{\sigma}$ by exactness assumption
  4. $\implies \tau(\kappa(\tau(\boldsymbol{\sigma}))) = \tau(\boldsymbol{\sigma})$ because $\tau$ is a function
  5. $\implies \tau(\kappa(\boldsymbol{\delta})) = \boldsymbol{\delta}$ substitute $\tau(\boldsymbol{\sigma}) \mapsto \boldsymbol{\delta}$
  6. $\implies \varphi(\boldsymbol{\delta}) = \boldsymbol{\delta}$ by definition of $\varphi$
  7. Thus, this direction holds.
- **Part 2** ($\boldsymbol{\delta} \in \mathcal{D} \impliedby \varphi(\boldsymbol{\delta}) = \boldsymbol{\delta}$):
  1. Suppose $\varphi(\boldsymbol{\delta}) = \boldsymbol{\delta}$
  2. $\implies \boldsymbol{\delta} = \tau(\kappa(\boldsymbol{\delta}))$
  3. $\implies \exists \boldsymbol{\sigma} \in \Sigma^* : \boldsymbol{\delta} = \tau(\boldsymbol{\sigma})$; specifically, $\boldsymbol{\sigma} = \tau(\boldsymbol{\delta})$
  4. $\implies \boldsymbol{\delta} \in \mathcal{D}$ by definition of $\mathcal{D}$
  5. Thus, this direction holds.

Since each directions have been proven, the proposition holds. ∎

**Corollary 1.**

$$\mathcal{D} = \{\varphi(\boldsymbol{\delta}) \mid \boldsymbol{\delta} \in \Delta^*\} \tag{27}$$

**Proposition 7.** *Let $(\tau, \kappa)$ be exact. Then, $(\tau, \kappa)$ is a bijection between $(\Sigma^*, \mathcal{D})$.*

*Proof.* The bijection follows directly from the following:

- Exactness means $\forall \boldsymbol{\sigma} \in \Sigma^* : \kappa(\tau(\boldsymbol{\sigma})) = \boldsymbol{\sigma}$.
- Proposition 6 ensures that $\forall \boldsymbol{\delta} \in \mathcal{D} : \tau(\kappa(\boldsymbol{\delta})) = \boldsymbol{\delta}$.

Thus, we have a bijection, and the proposition holds. ∎

**Proposition 8.** *For an exact tokenization model $(\Sigma, \Delta, \tau, \kappa)$, the canonicalization operator is idempotent, i.e.,*

$$\varphi(\boldsymbol{\delta}) = \boldsymbol{\delta}' \implies \varphi(\boldsymbol{\delta}') = \boldsymbol{\delta}' \text{ for all } \boldsymbol{\delta}, \boldsymbol{\delta}' \in \Delta^* \tag{28}$$

*Proof.* The proposition is equivalent to $\varphi(\varphi(\boldsymbol{\delta})) = \varphi(\boldsymbol{\delta})$ for all $\boldsymbol{\delta} \in \Delta^*$.

Fix $\boldsymbol{\delta}, \boldsymbol{\delta}' \in \Delta^*$ arbitrarily.

1. Suppose $\varphi(\boldsymbol{\delta}) = \boldsymbol{\delta}'$
2. $\implies \tau(\kappa(\boldsymbol{\delta})) = \boldsymbol{\delta}'$ by definition of $\varphi$
3. $\implies \tau(\kappa(\tau(\kappa(\boldsymbol{\delta})))) = \tau(\kappa(\boldsymbol{\delta}'))$ because $\tau$ and $\kappa$ are functions
4. $\implies \tau(\kappa(\tau(\kappa(\boldsymbol{\delta})))) = \tau(\kappa(\boldsymbol{\delta}'))$ by exactness
5. $\implies \tau(\kappa(\boldsymbol{\delta})) = \tau(\kappa(\boldsymbol{\delta}'))$
6. $\implies \tau(\kappa(\boldsymbol{\delta})) = \tau(\kappa(\varphi(\boldsymbol{\delta})))$ substitution $\boldsymbol{\delta}' \mapsto \varphi(\boldsymbol{\delta})$
7. $\implies \varphi(\boldsymbol{\delta}) = \varphi(\varphi(\boldsymbol{\delta}))$ by definition of $\varphi$

∎

## D. Proofs for Section 4 (Canonicality by Conditioning)

**Proposition 1.** *Assuming that the true distribution over tokens $p_\Delta^\star$ is canonical, the globally canonicalized model $g$ guarantees the following reduction in KL divergence:*

$$\mathrm{KL}(p_\Delta^\star \| p_\Delta) - \mathrm{KL}(p_\Delta^\star \| g) = \underbrace{-\log Z}_{\geq 0} \tag{7}$$

*Proof.*

$$\mathrm{KL}(p_\Delta^\star \| g) = \sum_{\boldsymbol{\delta} \in \Delta^*} p_\Delta^\star(\boldsymbol{\delta}) \log \frac{p_\Delta^\star(\boldsymbol{\delta})}{g(\boldsymbol{\delta})} \tag{29a}$$

$$= -\mathrm{H}(p_\Delta^\star) - \sum_{\boldsymbol{\delta} \in \Delta^*} p_\Delta^\star(\boldsymbol{\delta}) \log \frac{p_\Delta(\boldsymbol{\delta}) \mathbb{1}\{\boldsymbol{\delta} \in \mathcal{D}\}}{Z} \tag{29b}$$

$$= -\mathrm{H}(p_\Delta^\star) - \sum_{\boldsymbol{\delta} \in \Delta^*} p_\Delta^\star(\boldsymbol{\delta}) \log p_\Delta(\boldsymbol{\delta}) - \overbrace{\sum_{\boldsymbol{\delta} \in \Delta^*} p_\Delta^\star(\boldsymbol{\delta}) \log \mathbb{1}\{\boldsymbol{\delta} \in \mathcal{D}\}}^{0} + \sum_{\boldsymbol{\delta} \in \Delta^*} p_\Delta^\star(\boldsymbol{\delta}) \log Z \tag{29c}$$

$$= -\mathrm{H}(p_\Delta^\star) - \sum_{\boldsymbol{\delta} \in \Delta^*} p_\Delta^\star(\boldsymbol{\delta}) \log p_\Delta(\boldsymbol{\delta}) + \log Z \tag{29d}$$

$$= \mathrm{KL}(p_\Delta^\star \| p_\Delta) + \log Z \tag{29e}$$

Now, the proposition follows by basic algebra. ∎

**Proposition 2.**

$$\ell(\boldsymbol{\delta}) = p_\Delta(\boldsymbol{\delta}) \mathbb{1}\{\boldsymbol{\delta} \in \mathcal{D}\} \underbrace{\left[ \prod_{t=1}^{|\boldsymbol{\delta}|+1} \overrightarrow{\ell}(\boldsymbol{\delta}_{<t}) \right]^{-1}}_{\overset{\text{def}}{=} w_\ell(\boldsymbol{\delta})} \tag{10a}$$

*Thus,*

$$w_\ell(\boldsymbol{\delta}) = \frac{p_\Delta(\boldsymbol{\delta})}{\ell(\boldsymbol{\delta})} \mathbb{1}\{\boldsymbol{\delta} \in \mathcal{D}\} \tag{10b}$$

*Proof.*

$$\ell(\boldsymbol{\delta}) = \overrightarrow{\ell}(\text{EOS} \mid \boldsymbol{\delta}) \prod_{t=1}^{|\boldsymbol{\delta}|} \overrightarrow{\ell}(\delta_t \mid \boldsymbol{\delta}_{<t}) \tag{30a}$$

$$= \frac{p_\Delta(\text{EOS} \mid \boldsymbol{\delta}) \mathbb{1}\{\boldsymbol{\delta} \in \mathcal{D}\}}{\overrightarrow{\ell}(\boldsymbol{\delta})} \prod_{t=1}^{|\boldsymbol{\delta}|} \frac{p_\Delta(\delta_t \mid \boldsymbol{\delta}_{<t}) \mathbb{1}\{\boldsymbol{\delta}_{<t} \cdot \delta_t \in \overrightarrow{\mathcal{D}}\}}{\overrightarrow{\ell}(\boldsymbol{\delta}_{<t})} \tag{30b}$$

$$= \left[ p_\Delta(\text{EOS} \mid \boldsymbol{\delta}) \prod_{t=1}^{|\boldsymbol{\delta}|} p_\Delta(\delta_t \mid \boldsymbol{\delta}_{<t}) \right] \left[ \mathbb{1}\{\boldsymbol{\delta} \in \mathcal{D}\} \prod_{t=1}^{|\boldsymbol{\delta}|} \mathbb{1}\{\boldsymbol{\delta}_{<t} \cdot \delta_t \in \overrightarrow{\mathcal{D}}\} \right] \left[ \prod_{t=1}^{|\boldsymbol{\delta}|+1} \overrightarrow{\ell}(\boldsymbol{\delta}_{<t}) \right]^{-1} \tag{30c}$$

$$= p_\Delta(\boldsymbol{\delta}) \mathbb{1}\{\boldsymbol{\delta} \in \mathcal{D}\} \left[ \prod_{t=1}^{|\boldsymbol{\delta}|+1} \overrightarrow{\ell}(\boldsymbol{\delta}_{<t}) \right]^{-1} \tag{30d}$$

∎

**Proposition 3.** *Assuming that the true distribution over tokens $p_\Delta^\star$ is canonical, the locally canonicalized model $\ell$ guarantees the following reduction in KL divergence:*

$$\mathrm{KL}(p_\Delta^\star \| p_\Delta) - \mathrm{KL}(p_\Delta^\star \| \ell) = \underbrace{- \mathop{\mathbb{E}}_{\boldsymbol{\delta} \sim p_\Delta^\star} [\log w_\ell(\boldsymbol{\delta})]}_{\geq 0} \tag{11}$$

*Proof.*

$$\text{KL}(p_\Delta^\star \,\|\, \ell) = \sum_{\delta \in \Delta^*} p_\Delta^\star(\delta) \log \frac{p_\Delta^\star(\delta)}{\ell(\delta)} \tag{31a}$$

$$= -\text{H}(p_\Delta^\star) - \sum_{\delta \in \Delta^*} p_\Delta^\star(\delta) \log \ell(\delta) \tag{31b}$$

$$= -\text{H}(p_\Delta^\star) - \sum_{\delta \in \Delta^*} p_\Delta^\star(\delta) \log \underbrace{\frac{p_\Delta(\delta)\mathbb{1}\{\delta \in \mathcal{D}\}}{w_\ell(\delta)}}_{\text{by Proposition 2}} \tag{31c}$$

$$= -\text{H}(p_\Delta^\star) - \sum_{\delta \in \Delta^*} p_\Delta^\star(\delta) \log p_\Delta(\delta) - \overbrace{\sum_{\delta \in \Delta^*} p_\Delta^\star(\delta) \log \mathbb{1}\{\delta \in \mathcal{D}\}}^{\text{0 by assumption}} + \sum_{\delta \in \Delta^*} p_\Delta^\star(\delta) \log w_\ell(\delta) \tag{31d}$$

$$= \text{KL}(p_\Delta^\star \,\|\, p_\Delta) + \mathop{\mathbb{E}}_{\delta \sim p_\Delta^\star} \left[ \log w_\ell(\delta) \right] \tag{31e}$$

Now, the proposition follows by algebra. ∎

**Proposition 4.**

$$Z = \mathop{\mathbb{E}}_{\delta \sim \ell}\left[w_\ell(\delta)\right] = \mathop{\mathbb{E}}_{\delta \sim \ell}\left[\frac{p_\Delta(\delta)\mathbb{1}\{\delta \in \mathcal{D}\}}{\ell(\delta)}\right] \tag{12}$$

*Proof.*

$$Z = \sum_{\delta \in \Delta^*} p_\Delta(\delta)\mathbb{1}\{\delta \in \mathcal{D}\} \tag{32a}$$

$$= \sum_{\delta \in \mathcal{D}} p_\Delta(\delta) \tag{32b}$$

$$= \sum_{\delta \in \mathcal{D}} \ell(\delta)\frac{p_\Delta(\delta)}{\ell(\delta)} \tag{32c}$$

$$= \sum_{\delta \in \mathcal{D}} \ell(\delta)\frac{p_\Delta(\delta)}{p_\Delta(\delta)w_\ell(\delta)^{-1}} \tag{32d}$$

$$= \sum_{\delta \in \mathcal{D}} \ell(\delta)\, w_\ell(\delta) \tag{32e}$$

$$= \mathop{\mathbb{E}}_{\delta \sim \ell}\left[w_\ell(\delta)\right] \tag{32f}$$

∎

# E. Frequently Generated Noncanonical Bigrams

Because BPE canonicality is determined entirely by bigram constraints (App. B), a natural question is: *Which of those constraints does a language model violate most often?* To find out, we catalogue the model's most frequently produced noncanonical bigrams in hopes of revealing some specific aspects of BPE canonicality that it struggles to model.

The **bigram frequency** is defined as

$$\mu(\alpha{\cdot}\beta) \stackrel{\text{def}}{=} \mathop{\mathbb{E}}_{\boldsymbol{\delta}\sim p_\triangle}\left[\sum_{\delta\cdot\delta'\in\text{BIGRAMS}(\boldsymbol{\delta})}\mathbb{1}\{\delta{\cdot}\delta' = \alpha{\cdot}\beta\}\right] = \mathop{\mathbb{E}}_{\boldsymbol{\delta}\sim p_\triangle}\left[\sum_{t=1}^{|\boldsymbol{\delta}|-1}\mathbb{1}\{\delta_t{\cdot}\delta_{t+1} = \alpha{\cdot}\beta\}\right] \tag{33}$$

We must estimate $\mu$ because computing $\mu$ exactly is infeasible. We do so by sampling: Given a sample $\boldsymbol{\delta}^{(1)}, \dots, \boldsymbol{\delta}^{(M)} \stackrel{\text{i.i.d.}}{\sim} p_\triangle$, the following is an (unbiased) **Monte Carlo estimator**:

$$\widetilde{\mu}(\alpha{\cdot}\beta) \stackrel{\text{def}}{=} \frac{1}{M}\sum_{m=1}^{M}\sum_{t=1}^{|\boldsymbol{\delta}^{(m)}|-1}\mathbb{1}\{\delta_t^{(m)}{\cdot}\delta_{t+1}^{(m)} = \alpha{\cdot}\beta\} \tag{34}$$

However, we found that using the following (unbiased) **Rao–Blackwellized Monte Carlo estimator** provided better coverage for noncanonical bigrams, as they are individually rare:

$$\widehat{\mu}(\alpha{\cdot}\beta) \stackrel{\text{def}}{=} \frac{1}{M}\sum_{m=1}^{M}\sum_{t=1}^{|\boldsymbol{\delta}^{(m)}|-1}\sum_{\delta'\in\triangle}\mathbb{1}\{\delta_t^{(m)}{\cdot}\delta' = \alpha{\cdot}\beta\}\overrightarrow{p_\triangle}(\delta' \mid \boldsymbol{\delta}_{<t+1}^{(m)}) \tag{35}$$

The Rao–Blackwellized estimator uses the exact conditional expectation over the token at step $t + 1$ rather than the single sample token $\delta_{t+1}^{(m)}$, providing a significant variance reduction with little effect on the running time.

**Experiment.** Tab. 1 and 2 show the noncanonical bigrams (i.e., $\alpha{\cdot}\beta \notin \mathcal{B}$) that we have estimated (for each model) to be the most frequent via $\widetilde{\mu}$ with $M = 2000$ samples.

**Observations and takeaways.** Inspecting the tables, we see that a large portion of the frequently generated noncanonical bigrams are frequent substrings that have been tokenized incorrectly, for example, $\underset{3487}{\text{\_normal}}\cdot\underset{1143}{\text{ized}}$ corresponds to a frequent substring $\text{\_normalized}$ which—all things equal—should be tokenized as $\underset{39279}{\text{\_normalized}}$. However, there does not appear to be a clear pattern for why these models struggle with these specific bigrams over others; further investigation is needed.

| GPT-2 Small | | GPT-2 Medium | | GPT-2 Large | |
|---|---|---|---|---|---|
| **Bigram** | **Freq.** | **Bigram** | **Freq.** | **Bigram** | **Freq.** |
| ri eros (380, 27498) | 3.86e-3 | ␣ \xc2\xa0 (220, 1849) | 4.58e-3 | ␣ \xc2\xa0 (220, 1849) | 4.67e-3 |
| ␣NG PF (39058, 42668) | 1.99e-3 | neck er (27235, 263) | 2.36e-3 | sson i (16528, 72) | 2.81e-3 |
| Be ech (3856, 3055) | 1.94e-3 | h ands (71, 1746) | 2.33e-3 | ␣SA E (14719, 36) | 1.96e-3 |
| t kB (83, 38841) | 1.66e-3 | Che w (7376, 86) | 1.73e-3 | ␣Bul g (8510, 70) | 1.71e-3 |
| \\ \\\\ (59, 6852) | 1.59e-3 | ' ,' (11, 11) | 8.73e-4 | ) \xe2\x80\x94 (8, 960) | 1.36e-3 |
| ␣Red uce (2297, 7234) | 1.56e-3 | US A (2937, 32) | 8.10e-4 | ␣Ans ys (28038, 893) | 1.18e-3 |
| D ub (35, 549) | 1.35e-3 | go ers (2188, 364) | 7.68e-4 | upp ing (7211, 278) | 6.24e-4 |
| python y (29412, 88) | 1.21e-3 | st rush (301, 37357) | 6.81e-4 | ' ,' (11, 11) | 5.93e-4 |
| ␣sp t (599, 83) | 1.20e-3 | ␣coll ide (2927, 485) | 5.91e-4 | ␣J udd (449, 4185) | 5.32e-4 |
| ␣inspect or (10104, 273) | 9.64e-4 | ␣Black hawks (2619, 27221) | 5.82e-4 | ␣polite ly (23507, 306) | 4.84e-4 |
| EM P (3620, 47) | 9.56e-4 | unt r (2797, 81) | 4.52e-4 | ing o (278, 78) | 4.08e-4 |
| EF OR (25425, 1581) | 9.34e-4 | med iated (1150, 12931) | 4.31e-4 | stan bul (14192, 15065) | 3.97e-4 |
| II T (3978, 51) | 7.36e-4 | ␣ce iling (2906, 4386) | 4.01e-4 | === - (18604, 12) | 3.97e-4 |
| ␣normal ized (3487, 1143) | 5.78e-4 | og ging (519, 2667) | 3.75e-4 | ␣cancel ed (14241, 276) | 3.92e-4 |
| ␣Gen ocide (5215, 16207) | 5.34e-4 | D un (35, 403) | 3.11e-4 | ␣Congress man (3162, 805) | 3.44e-4 |
| ␣V og (569, 519) | 4.70e-4 | ␣Bur st (5481, 301) | 2.41e-4 | ␣ESP N (9428, 45) | 3.43e-4 |
| ␣wrong ful (2642, 913) | 4.46e-4 | oxin emia (39366, 22859) | 2.39e-4 | ski pped (20545, 1496) | 2.88e-4 |
| ab us (397, 385) | 3.56e-4 | Che wan (7376, 8149) | 2.27e-4 | ␣eat en (4483, 268) | 2.62e-4 |
| ␣An nie (1052, 11952) | 3.46e-4 | qt x (39568, 87) | 2.27e-4 | ␣repair ing (9185, 278) | 2.51e-4 |
| ␣provision al (8287, 282) | 2.82e-4 | Che wich (7376, 11451) | 2.26e-4 | m ology (76, 1435) | 2.29e-4 |

*Table 1.* Most frequently generated noncanonical bigrams generated by GPT-2 models.

| Llama 3.2-1B | | Llama 3.2-3B | | Llama 3.1-8B | |
|---|---|---|---|---|---|
| **Bigram** | **Freq.** | **Bigram** | **Freq.** | **Bigram** | **Freq.** |
| \xc2\xa0\xc2\xa0 \xc23\xa0 
 9421 · 4194 | 5.15e-3 | \xe3\x80\x80 \xe3\x80\x80 
 23249 · 23249 | 6.41e-3 | \xe3\x80\x80 \xe3\x80\x80 
 23249 · 23249 | 9.60e-3 |
| \xc2\xa0 \xc2\xa0 
 4194 · 4194 | 3.06e-3 | \xc2\xa0 \xc2\xa0 
 107958 · 4194 | 4.64e-3 | ␣Rivers ide 
 41416 · 3633 | 3.46e-3 |
| \t \t 
 197 · 197 | 2.04e-3 | + y 
 10 · 88 | 1.86e-3 | ␣anthrop ony 
 41416 · 3633 | 1.13e-3 |
| .c rt 
 522 · 3423 | 6.67e-4 | ende w 
 11178 · 86 | 1.60e-3 | ol ta 
 337 · 2629 | 6.92e-4 |
| UE ST 
 2279 · 790 | 6.33e-4 | ␣((( sqrt 
 11861 · 27986 | 1.54e-3 | ␣NI K 
 42601 · 42 | 4.55e-4 |
| ␣UP LOAD 
 12250 · 13285 | 2.60e-4 | ␣ro bert 
 938 · 9339 | 1.07e-3 | \xc2\xa0 \xc2\xa0 
 4194 · 4194 | 4.22e-4 |
| cole an 
 56745 · 276 | 2.47e-4 | \xc2\xa0 ␣ 
 107958 · 220 | 9.16e-4 | \xc2\xa0 ␣ 
 4194 · 220 | 3.27e-4 |
| ␣Ind i 
 2314 · 72 | 2.10e-4 | \xc2\xa0 \xc2\xa0 
 4194 · 4194 | 5.75e-4 | KA YA 
 27542 · 87582 | 3.05e-4 |
| ␣UP DATE 
 12250 · 7242 | 2.03e-4 | set t 
 751 · 83 | 5.50e-4 | KA YY 
 27542 · 10266 | 2.70e-4 |
| F ERIC 
 37 · 37016 | 2.03e-4 | \xc2\xa0 ␣ 
 17529 · 220 | 3.52e-4 | ␣membr anes 
 31819 · 14997 | 2.67e-4 |
| vol ved 
 12821 · 2111 | 2.03e-4 | nam en 
 12682 · 268 | 3.49e-4 | ␣Pro phet 
 1322 · 22718 | 2.25e-4 |
| cole ans 
 56745 · 598 | 2.00e-4 | + x 
 10 · 87 | 2.32e-4 | And y 
 3112 · 88 | 1.67e-4 |
| F ERICA 
 37 · 96390 | 1.93e-4 | \xc2\xa0 \xc2\xa0 
 17529 · 4194 | 2.29e-4 | og ens 
 540 · 729 | 1.37e-4 |
| arm ae 
 2227 · 6043 | 1.60e-4 | ,M ovie 
 28112 · 6461 | 1.62e-4 | \xd0\xb2\xd0\xb0\xd0\xbd \xd1\x96 
 104471 · 27385 | 1.29e-4 |
| F ORIA 
 37 · 90118 | 1.53e-4 | ip les 
 575 · 645 | 1.11e-4 | ␣Sou ls 
 9424 · 4835 | 1.29e-4 |
| ␣SER IES 
 14311 · 5482 | 1.43e-4 | ␣ReSharper ed 
 53255 · 291 | 1.07e-4 | ␣anthrop a 
 41416 · 64 | 1.11e-4 |
| M OST 
 44 · 4327 | 1.20e-4 | atty t 
 23758 · 83 | 1.07e-4 | GE l 
 11010 · 75 | 1.03e-4 |
| ␣beg ins 
 2197 · 1354 | 1.13e-4 | ␣␣ collections 
 721 · 52237 | 1.03e-4 | ␣I van 
 358 · 16023 | 9.89e-5 |
| - Serve 
 12 · 61521 | 9.67e-5 | ip ple 
 575 · 698 | 1.01e-4 | ␣diff usion 
 3722 · 7713 | 9.33e-5 |
| IR ON 
 2871 · 715 | 9.33e-5 | ␣G hana 
 480 · 84203 | 1.01e-4 | ACC ORD 
 30542 · 4373 | 9.01e-5 |

*Table 2.* Most frequently generated noncanonical bigrams generated by Llama models.

