# OpenReview forum: "Language Models over Canonical Byte-Pair Encodings"
_ICML.cc/2025/Conference — ICML 2025 poster_

### Official Review · Reviewer_1h8t · 2025-03-14

**Overall Recommendation:** 4

**Summary:**

The submission discusses issues arising from tokenization wherein language models place positive probability mass on sequences unobservable during training. The submission presents approaches for both test time and train time for lessening the severity of this issue.

**Claims And Evidence:**

> Are the claims made in the submission supported by clear and convincing evidence?

Yes, the paper is quite well written up until the experiments section.

**Essential References Not Discussed:**

> Are there related works that are essential to understanding the (context for) key contributions of the paper, but are not currently cited/discussed in the paper?

No.

**Experimental Designs Or Analyses:**

The experimental design seems fine.

**Methods And Evaluation Criteria:**

GPT2 is a bit outdated, but it is enough to demonstrate the claim of the paper.

**Other Comments Or Suggestions:**

The submission introduces $\cdot$ for concatenation but then switches to $\langle \rangle$ for bigrams.

Byte-rair encoding

In Section 5, I find the bulleting without quite jarring to read.

It would help clarity to remind the reader of definitions in more places (the submission uses a lot of different symbols and symbol modifiers).

**Other Strengths And Weaknesses:**

The main idea of the paper is very simple, but the conceptual clarity of the paper is impressive nonetheless. The writing could do a bit more handholding in the methodology and experiment sections, but even these exceed the quality of a typical machine learning paper. I'm sure local canonicalization exists in some codebases, but I haven't seen it nicely formalized in literature. The idea of training or fine-tuning in a way that enforces canonicalization feels like it has the potentially to be quite useful.

**Questions For Authors:**

None.

**Relation To Broader Scientific Literature:**

The submission does a sufficient job relating the paper to broader scientific literature.

**Theoretical Claims:**

I did not check the proofs.

---

> ### Author Rebuttal · Authors · 2025-04-01
>
> ### General Response
>
> **Clarity and notation improvements**: Several reviewers pointed out that our mathematical presentation, although precise, could be more reader-friendly in various places. In the next revision, we will include more reminders about notation and add several clarifying remarks where beneficial.
>
> **Experimental setup and discussion**: We recognize that many of the reviewers found our experiments section excessively terse. We completely agree and will seek to improve the readability of this section in the next revision, which will be easier with the additional space.
>
> **Evaluation of downstream accuracy**: Our initial submission focuses on a theoretical characterization of the problem and our proposed solutions. As such, our evaluation was mainly focused on the task-agnostic measures of (a) the rate of canonicality in language models and (b) the effectiveness of our proposed algorithms for enforcing canonicality. However, we agree with the reviewers that including some evaluation of the downstream effects of our canonicalization methods will considerably strengthen the paper.  We will investigate the feasibility of extending our evaluation to include some common LM reasoning benchmarks, e.g., HellaSWAG and GLUE.  If the reviewers have other/better suggestions, we kindly ask that they share them.
>
>
> ### Response to Reviewer 1h8t
>
> Thank you for your review!  We are sorry about the "jarring" terseness of the experiment section. We will revise it in the next revision.
>
> > GPT2 is a bit outdated, but it is enough to demonstrate the claim of the paper.
>
> Note that we included experiments with Llama 3.2-1B as well as GPT-2-[small, medium, large]. In that setting, we saw that Llama 3.2-1B's canonicality rate was surprisingly significantly lower (.763) than the GPT-2 models ([0.895, 0.929, 0.944]).  This means that the more modern Llama model benefits MORE from conditionally canonical generation.
>
> Nonetheless, we agree that our experiments could benefit from a wider range of language models. For the next revision, we will extend our evaluation to include some of the larger Llama models.
>
> In the fine-tuning experiments, we limited them ot GPT-2-small and medium.  Since these experiments are the most compute intensive.
>
> > The writing could do a bit more handholding in the methodology and experiment sections, but even these exceed the quality of a typical machine learning paper.
>
> We will revise the paper to include more "handholding" and reminders to help unpack the terse notation.
>
> See also: general response
>
> > I'm sure local canonicalization exists in some codebases, but I haven't seen it nicely formalized in literature.
>
> The closest methods we could find are discussed in appendix B.3.  As noted in appendix B.3, our method is significantly simpler to implement and uses significantly less memory.  In fact, we initially tried using these methods for this paper, but they consumed so much memory that we couldn't build the complete automaton without running out of memory.
>
> We suspect that if other canonicality filtering methods exist in other code bases, they are heuristics rather than principled methods with proofs.  Getting the details right about the BPE canonicality filter was surprisingly nontrivial; as you can see, we have an extensive appendix (B) detailing why our method is correct.  However, we note that the final filtering rule that we derived through all that painstakingly detailed work is very simple to implement (a simple bigram test) and it is possible that someone out there "intuited" it, but if they had proven it correct, we'd strongly suspect that they would have at least arXived it or released code for it unless proprietary :-(
>
> > The submission introduces ⋅ for concatenation but then switches to ⟨⟩ for bigrams.
>
> Good point - we could use juxtapositions to denote the bigram instead of
>
> > Byte-rair encoding
>
> Sorry about that - we spotted that one after submission!
>
> > In Section 5, I find the bulleting without quite jarring to read.
>
> We will improve the readability.  In particular, the terse style of the experiment section in the final version.  It was written in this very terse style because we were short on space.  We appreciate you for pushing through despite the jarring style!
>
> > It would help clarity to remind the reader of definitions in more places (the submission uses a lot of different symbols and symbol modifiers).
>
> Thank you - we agree that adding gentle reminders about the notation throughout the paper will improve the readability a lot, and we will do so in our next revision.

---

### Official Review · Reviewer_y53f · 2025-03-15

**Overall Recommendation:** 4

**Summary:**

This paper considers an issue with language models trained on BPE-tokenized sequences, where they assign positive probability to so-called non-canonical sequences that could not result from the BPE encoding procedure. They give efficient membership tests for canonicality, and several methods for enforcing the support of the model to be contained in the canonical sequences; one set of methods is through more expensive inference time sampling algorithms, and one is via the parameterization of the language model (which uses additional training). Experiments show that these methods can indeed significantly increase the modelled likelihood across several datasets.

## Update after rebuttal

The authors have answered well some questions and clarifications I had. Including some of the discussion into new versions of the paper could make the paper even better (its state at submission was already very novel and interesting). I recommend acceptance.

**Claims And Evidence:**

In general, the claims are supported.

**Essential References Not Discussed:**

n/a

**Experimental Designs Or Analyses:**

In my opinion, there are insufficient experimental details, as I mentioned above.

**Methods And Evaluation Criteria:**

For the experiment in Section 5.3, the improvement in perplexity of $l_\theta$ is conflated with the fact that $l_\theta$ is trained partially with the language modelling loss on the dataset of interest. A useful baseline would be to train $p_{\Delta}$ with the language modelling loss on PTB and WikiText-103, without any canonicality constraints. Some additional experimental details here would also be helpful such as hyperparameters and training setup. Also, if the perplexity reported on a test set, or the same data that it was trained on?

**Other Comments Or Suggestions:**

$\preceq$ symbol is undefined. This was confusing to me.

Top left table on page 8 should be labelled with $-\log(\hat{Z})$.

**Other Strengths And Weaknesses:**

This paper elegantly tackles a systematic issue in a widely used technique. It seems pretty novel, and it has theoretical insights that connect closely to actually empirical practice. For instance, in Appendix A, the authors include thoughtful comments and examples related to BPE and tokenization in actual language models. However, I do have some questions about the utility of the work (see below).

**Questions For Authors:**

In the "Why train?" section. Suppose we initialized a standard (non-canonicalized) language model at $\theta^{(0)}$ such that the support is on canonical sequences only. If trained with say maximum likelihood on only canonical sequences, would it be the case that this standard model also has no optimization pressure for canonicality, because the gradients will not assign mass to non-canonical sequences?

I have trouble understanding how misallocated probability mass could impact practical applications of LLMs. If all sequences had a constant bias in probability caused by mass on noncanonical sequences, then this would have no impact, correct? Perhaps if there were systematic biases, say in a domain like mathematical proofs, where mass on noncanonical sequences leads to higher probabilities of wrong answers, then this could be an issue. Any thoughts on this?

Another related question: could the ability to assign positive probability to certain sequences be beneficial in certain cases? For instance, if the language model makes a mistake and starts generating a canonical sequence of tokens that leads to a wrong answer, perhaps it may have to start generating a noncanonical sequence to get to a right answer? Or, perhaps the user prompt for a base language model is cut-off or has a typo at the end, and to complete the sentence it may be relatively easy for a language model to complete the query with a noncanonical sequence of tokens? E.g. if the prompt were "My GPUs are from NVI", and a good completion is "My GPUs are from NVIDIA", and NVIDIA were a token, might it be hard for a canonicalized language model to output the desired completion?

**Relation To Broader Scientific Literature:**

BPE is an extremely common method for LLM tokenizers, which has several noted downsides. Studies that improve BPE or work towards different tokenization frameworks could be very impactful.

**Theoretical Claims:**

In the proofs of proposition 1 and 3, the term $H(p^*_{\Delta})$ is missing a negative sign throughout, but this is a small issue.

I did not check most other proofs.

---

> ### Author Rebuttal · Authors · 2025-04-01
>
> **Clarity and notation improvements**: Several reviewers pointed out that our mathematical presentation, although precise, could be more reader-friendly in various places. In the next revision, we will include more reminders about notation and add several clarifying remarks where beneficial.
>
> **Experimental setup and discussion**: We recognize that many of the reviewers found our experiments section excessively terse. We completely agree and will seek to improve the readability of this section in the next revision, which will be easier with the additional space.
>
> **Evaluation of downstream accuracy**: Our initial submission focuses on a theoretical characterization of the problem and our proposed solutions. As such, our evaluation was mainly focused on the task-agnostic measures of (a) the rate of canonicality in language models and (b) the effectiveness of our proposed algorithms for enforcing canonicality. However, we agree with the reviewers that including some evaluation of the downstream effects of our canonicalization methods will considerably strengthen the paper.  We will investigate the feasibility of extending our evaluation to include some common LM reasoning benchmarks, e.g., HellaSWAG and GLUE.  If the reviewers have other/better suggestions, we kindly ask that they share them.
>
> ### Response to y53f
>
> Thank you for your review, suggestions, and interesting questions!
>
> > Methods And Evaluation Criteria
>
> Fantastic suggestion - I can't believe we missed that!
>
> The perplexity evaluation is on a held-out test set.  We will clarify this in the next revision.
>
> We will add more details about the experimental setup and release the code upon publication.
>
> > Missing minus sign in Prop 1 and 3
>
> Thanks for catching that!
>
> > In my opinion, there are insufficient experimental details, as I mentioned above.
>
> Please see the general response.
>
> > ⪯ symbol is undefined
>
> Sorry about that! x ⪯ y it means x is a prefix of y
>
> > Top left table on page 8
>
> Thanks!
>
> > In the "Why train?" section. Suppose we initialized a standard (non-canonicalized) language model at θ(0) such that the support is on canonical sequences only.
>
> Softmax always assigns a nonzero probability to all options.  So it is not possible unless we change the model's parameterization to not use softmax.
>
> > If trained with say maximum likelihood on only canonical sequences, would it be the case that this standard model also has no optimization pressure for canonicality, because the gradients will not assign mass to non-canonical sequences?
>
> Intuition: At the output layer, the gradient update does two things: (1) it attempts to increase the probability of the (observed) canonical sequences, and (2) it attempts to decrease the probability of all unobserved sequences (including the noncanonical sequences).
>
> However, because the network has many layers, pushing the probability of the observed sequence up may have the unwanted side effect of increasing the probability of some noncanonical tokenizations that share latent representations, etc.  Note that the gradient signal also seeks to lower the probability of unobserved sequences (which include noncanonical sequences). This means that the latent representations in the network will be updated to push those probabilities down.
>
> In our revised parameterization, the network does not have to learn the canonicality function - it will be baked into the model.  So, during learning, the gradient update can focus on teasing apart the plausible candidates, not the ones that we can rule out with our filter.  This will likely give a signal to learn from as the noise of needing to push down the probability of noncanonical sequences is removed.
>
> > understanding misallocated probability
>
> Indeed, the main concern is that noncanonical mass is allocated in ways that are not constant. That said, even if the mass allocated was a (small) constant to each noncanonical token, then ancestral sampling from the model would sample these tokens and the model will end up with essentially out-of-distrbution contexts. Additionally, the longer the model generates, the higher the probability of falling off the canonical path becomes.
>
> > "My GPUs are from NVI"
>
> This is known as the "prompt boundary problem," and it has been solved in [Vieira et al (2024)](https://arxiv.org/abs/2412.03719). They solve it by correctly conditioning the LM on a character string prefix.  We believe it is directly compatible with our canonicalized LM.  This is a much more direct approach than hoping that subword tokens happen to have picked up a [weak] signal that shouldn't be in the model to begin with unless the LM was trained with stochastic tokenizers or other kinds of data augmentation!

---

### Official Review · Reviewer_SszZ · 2025-03-17

**Overall Recommendation:** 4

**Summary:**

This paper addresses the problem of non-canonical tokenization, which arises when there are multiple tokenizations that decode to the same sequence of input characters (all of which are assigned some probability by a language model), but only one is ever produced by the deterministic tokenizer. The paper asserts that this is a problem, because it prevents the language model from properly learning the ground-truth distribution over sequences of tokens. They present two approaches for alleviating non-canonical tokenization. The first is a test-time strategy which does not require any extra training and generates text conditional on the canonically constraint, while the second requires training to ensure canonical tokenization outputs.

**Claims And Evidence:**

L140: “The tokenization is used in practice because modeling (short) token strings is easier than modeling (long)
character strings. Commonly used tokenizers—in particular, those based on byte-pair encoding (BPE; Sennrich et al., 2016; Gage, 1994b)—have this property, which is why they are widely used.” Is there a citation or evidence for this claim?

**Essential References Not Discussed:**

No

**Experimental Designs Or Analyses:**

Yes, they seemed reasonable at a high level

**Methods And Evaluation Criteria:**

Perplexity makes sense as a first evaluation, but additional evaluation datasets — which get at the quality of the outputs, not just the perplexity (so e.g. in-context learning tasks, math tasks, etc) — would strengthen the claim that canonicality is a genuine problem.

**Other Comments Or Suggestions:**

* The central claim of the paper is that non-canonicality is an obvious problem (“Any probability mass assigned to them is simply an error” L039). But a priori, this is not entirely clear (although it does seem very likely). Although language models are trained on next-token prediction, the evaluation we actually care about is at the word-level, so modeling the distribution over token sequences is not the ultimate goal — the ultimate goal is to generate high quality text. How can we be sure that

* The section in Appendix B that explains BPE could be made clearer in several ways, particularly with the inclusion of a detailed, step by step example labeled with each of the mathematically defined functions. It would also be helpful to repeat definitions from the main body, such as the definition of “BIGRAMS” or $\mathcal{B}$.

* I did not see where the notation for “is a prefix of” (squiggly less than or equal to symbol) was defined

* Where is g with a right arrow on top of it defined? If line 229, then would recommend to not use it before it is defined

* In Figure 2, the table contains token sequences, not strings? Also, what are the numbers (token IDs)?

* In the ancestral_sampling algorithm, L25, what is $\epsilon$?

* It is somewhat confusing to use $\ell$ for a language model, rather than for a loss function

* Explain somewhere what it means to have the arrow on top? Is there a consistent theme in all variables that have an arrow on top?

* Do 5.1 and 5.2 use the “conditioning” method, and 5.3 the “fine-tuning” method? This could be made more clear/explicit, particularly 5.1 and 5.2
* The table on the upper right of page 8 needs to have a caption

A few typos:

* Line 158, “the general tokenizers” -> “general tokenizers”
* Line 200, “distrbutions”
* Line 308, “distribution over” —> “distributions over”

**Other Strengths And Weaknesses:**

Strengths: The paper illuminates a widespread but overlooked phenomenon in the output of large language models, induced by tokenization. It appears to be an important problem, and they present multiple methods for addressing it which improve the perplexity in experiments, deriving useful results/algorithms relating to byte pair encoding along the way. The explanations are relatively clear (with the small exception of some notation, noted in the next section)

Weaknesses:

Evaluation: ultimately, we care about the learned distribution over characters, not token sequences. However, perplexity is the only metric evaluated, so we don’t actually know whether fixing the canonicalization problem really produces higher quality outputs. It would therefore strengthen the paper to evaluate their method on e.g. in-context learning, or question-answering, tasks.

Also, the paper’s experimental results find that the larger the model, the more likely it is to have learned to output canonical token sequences. This would seem to indicate that the marginal gain of canonicalizing could decrease with scale.

**Questions For Authors:**

1. In what sense is the second method a change in “model architecture”? The language models in Definition 3 look more like another test-time strategy (like the first method), which simply updates the sampling scheme. Am I misunderstanding something?
2. I don’t understand the paragraph “Why train”? (L330-L338). Which training objective for $\theta^{(0)}$ “pressured the parameters to modal canonicality”? What does “Thus, if the parameters are used to model canonicality preferences, they can be repurposed to model different phenomena.” mean — which phenomena?
3. Instead of encouraging, or enforcing, canonicality, what about using a randomized tokenization of the training data, i.e. making the tokenization method non-deterministic?
4. Recent work (e.g. the Byte Latent Tokenizer) has moved away from traditional tokenization methods like byte pair encoding. How could the ideas about canonicality in this paper apply to BLT, if at all?
5. Did the authors consider evaluating their method on any mathematical datasets? Is non-canonicality a problem for number tokenization?

**Relation To Broader Scientific Literature:**

The paper is very relevant to the LLM tokenization literature. However, it seems to make a very novel observation about canonicality, so there are not direct references that it clearly builds on. It also presents a new, simplified algorithm for detecting BPE canonicality.

**Theoretical Claims:**

No

---

> ### Author Rebuttal · Authors · 2025-04-01
>
> ### General Response
> **Clarity and notation improvements**: Several reviewers pointed out that our mathematical presentation, although precise, could be more reader-friendly in various places. In the next revision, we will include more reminders about notation and add several clarifying remarks where beneficial.
>
> **Experimental setup and discussion**: We recognize that many of the reviewers found our experiments section excessively terse. We completely agree and will seek to improve the readability of this section in the next revision, which will be easier with the additional space.
>
> **Evaluation of downstream accuracy**: Our initial submission focuses on a theoretical characterization of the problem and our proposed solutions. As such, our evaluation was mainly focused on the task-agnostic measures of (a) the rate of canonicality in language models and (b) the effectiveness of our proposed algorithms for enforcing canonicality. However, we agree with the reviewers that including some evaluation of the downstream effects of our canonicalization methods will considerably strengthen the paper.  We will investigate the feasibility of extending our evaluation to include some common LM reasoning benchmarks, e.g., HellaSWAG and GLUE.  If the reviewers have other/better suggestions, we kindly ask that they share them.
>
> ### Response to Reviewer SszZ
>
> > citation or evidence
>
> "Easier" is a bit open-ended here. One thing that is clearly "easier" is that training runs faster.  Some have conjectured that tokenization makes it "easier" to learn long-distance dependencies as the distance between tokens is shorter than the distance between (say) characters.  We will look for some citations to strengthen this claim, and we will rephrase it.
>
> > additional evaluation
>
> Please see the general response.
>
> > marginal decreases with scale
>
> Indeed, we did see in the case of GPT-2 (small, medium, large) that increasing model size/accuracy improves the canonicality rate.  However, we also saw in the case of Llama that the canonicality rate is very low.  We will include comparisons with the larger Llama models to see how the trend generalizes to the case of Llama.
>
> We also note that the longer the generated sequences are, the higher the noncanonicality rate becomes.  We will extend our canonicality rate experiments to include longer sequence lengths to better illustrate this point.
>
> > the ultimate goal
>
> Looks like you got cut off on that last sentence.  I will attempt to complete it in order to reply.  I suspect you were asking, "How can we be sure that canonicalizing the LM helps us generate high-quality text?"  This is a great question and one to dig into more in future work now that we have developed the machinery to answer it.  It is closely related to the general response about downstream evaluation.
>
> > Appendix B
>
> Great suggestion!
>
> > “is a prefix of”
>
> Sorry, for the oversight
>
> > Where is g with a right arrow on top of it defined? If line 229, then would recommend to not use it before it is defined
>
> Apologies! Eq. (11b) should have been a def=
>
> > Figure 2
>
> Yes, they are token sequences paired with their token ids.  The caption "The table shows short examples of canonical and noncanonical strings …" should be revised to say TOKEN strings.
>
> > ϵ
>
> ε is the length-0 string, defined in §2.1. We will add a reminder here, e.g., a pseudocode comment.
>
> > confusing to use ℓ
>
> We thought that ℓ for "local" and g for "global" made a good mnemonic. Do you have another suggestion? We use $\matcal{L}$ for the loss.
>
> > arrow on top
>
> We will explain our notational conventions. The arrow means that the function takes a prefix of a string rather than a complete string.
>
> > Do 5.1 and 5.2 use the “conditioning” method, and 5.3 the “fine-tuning” method? This could be made more clear/explicit, particularly 5.1 and 5.2
>
> Correct. We will improve the writing of section 5.
>
> > 1. model architecture
>
> The change: we take the transformer's output layer, which is a softmax over the entire vocabulary, and modify to sums over the canonical tokens given the prefix rather than all tokens.
>
> > 2. ... which phenomena?
>
> The "phenomena" we had in mind was *any* phenomena in language. So, the idea is that if the LM doesn't have to use its representational capacity to learn canonicality, then it could use its capacity to represent something else that helps it predict the next word better.
>
> > 3. non-deterministic tokenization
>
> Indeed, this is a possibility. There was a lot of interest in the early days of BPE in training with subword regularization, e.g., "BPE dropout" – the hope being that training on lots of different tokenization would provide a useful kind of data augmentation.  However, this idea is not used by any of the state-of-the-art large language models (see footnote 5).
>
> > 4. Byte Latent Tokenizer
>
> Unfortunately, we are not sufficiently familiar with this method to comment.
>
> > 5. additional evals
>
> See general response

---

### Official Review · Reviewer_99cZ · 2025-03-21

**Overall Recommendation:** 4

**Summary:**

This paper examines a key limitation of current language models: the fact that they allocate probability mass to token sequences that are impossible given their tokenizer. For example, for GPT4's tokenizer, the token sequence __t_, _he_ will never occur (since "the" is tokenized as __the_), yet GPT4 assigns a non-zero probability to it. The authors provide a formal characterization of this problem as "non-canonical" token encodings and introduce two approaches to enforce canonicality. They conduct a small series of experiments, finding that enforcing canonicality improves the likelihood of unseen data.

### Update after rebuttal

I continue to believe that this is a strong paper that merits acceptance, especially if the authors follow through on the commitments made during the rebuttal.

**Claims And Evidence:**

The claims in the paper are supported by convincing evidence: the theoretical analysis is sound as far as I can tell, and the experiments support the conclusions from the theoretical part.

**Essential References Not Discussed:**

There is not any _particular_ reference I am missing, but in general the paper could be better contextualized with respect to other papers that examine alternative tokenizations of the same character string. For example, there has been quite a bit of work examining the effects of non-canonical but morphologically correct tokenizations. The fact that all these studies have been dealing with a problem that only this paper characterizes in a theoretical way would further underscore the relevance of the findings.

[Different Tokenization Schemes Lead to Comparable Performance in Spanish Number Agreement](https://aclanthology.org/2024.sigmorphon-1.4/) (Arnett et al., SIGMORPHON 2024)

[Improving Tokenisation by Alternative Treatment of Spaces](https://aclanthology.org/2022.emnlp-main.786/) (Gow-Smith et al., EMNLP 2022)

[Superbizarre Is Not Superb: Derivational Morphology Improves BERT’s Interpretation of Complex Words](https://aclanthology.org/2021.acl-long.279/) (Hofmann et al., ACL 2021)

[Greed is All You Need: An Evaluation of Tokenizer Inference Methods](https://aclanthology.org/2024.acl-short.73/) (Uzan et al., ACL 2024)

**Experimental Designs Or Analyses:**

I checked the experimental analyses and could not detect any issues, apart from the fact that they are very short, and a more extensive evaluation, especially with respect to downstream tasks, would have been desirable (see above).

**Methods And Evaluation Criteria:**

The methods make sense, but the evaluation has certain limitations: it does not consider to what extent enforcing canonicality affects performance on downstream tasks. I think examining this would be important, as it would give an indication how valuable the paper's insights could be in practice.

**Other Comments Or Suggestions:**

You use two different citations for Gage (1994).

**Other Strengths And Weaknesses:**

I enjoyed reading this paper. It provides a rigorous theoretical treatment of a problem that has been floating around in the tokenization literature for a while; its insights could inspire a lot of follow-up work. The paper is also very well written -- I could not spot a single typo.

My main concern is about the missing breadth of the experimental evaluation, but this might still be acceptable for a paper whose focus is on a theoretical contribution.

**Questions For Authors:**

Is there any principled reason why you did not conduct any evaluation on downstream tasks?

**Relation To Broader Scientific Literature:**

The findings of the paper constitute a significant departure from prior work; the problem of alternative tokenizations of the same character string is well known in the literature, but the probabilistic treatment, as well as the proposed approaches for canonicalization, are novel as far as I can tell.

**Theoretical Claims:**

I checked the theoretical claims and could not detect any issues -- the analysis seems well-motivated and sound.

---

> ### Author Rebuttal · Authors · 2025-04-01
>
> ### General Response
> **Clarity and notation improvements**: Several reviewers pointed out that our mathematical presentation, although precise, could be more reader-friendly in various places. In the next revision, we will include more reminders about notation and add several clarifying remarks where beneficial.
>
> **Experimental setup and discussion**: We recognize that many of the reviewers found our experiments section excessively terse. We completely agree and will seek to improve the readability of this section in the next revision, which will be easier with the additional space.
>
> **Evaluation of downstream accuracy**: Our initial submission focuses on a theoretical characterization of the problem and our proposed solutions. As such, our evaluation was mainly focused on the task-agnostic measures of (a) the rate of canonicality in language models and (b) the effectiveness of our proposed algorithms for enforcing canonicality. However, we agree with the reviewers that including some evaluation of the downstream effects of our canonicalization methods will considerably strengthen the paper.  We will investigate the feasibility of extending our evaluation to include some common LM reasoning benchmarks, e.g., HellaSWAG and GLUE.  If the reviewers have other/better suggestions, we kindly ask that they share them.
>
>
> ### Response to Reviewer 99cZ
>
> Thank you for your review, kind words, and excellent suggestions for additional discussion and deepening our evaluations.
>
> > Essential References Not Discussed:
>
> Thanks for these suggestions. We will use them to contextualize our paper better in the broader tokenization literature.
>
> > Other Strengths And Weaknesses: I enjoyed reading this paper. It provides a rigorous theoretical treatment of a problem that has been floating around in the tokenization literature for a while; its insights could inspire a lot of follow-up work. The paper is also very well written -- I could not spot a single typo.
>
> Thank you!
>
> > My main concern is about the missing breadth of the experimental evaluation, but this might still be acceptable for a paper whose focus is on a theoretical contribution.
>
> Please see the general response.

---

### Decision · Program_Chairs · 2025-05-01

**Decision:**

Accept (poster)

**Comment:**

This paper points out that language models assign non-zero probabilities to token sequences which will never appear because of the tokenizer algorithm. The authors present two approaches to address this problem in language models, and show that addressing it increases the model's estimated likelihood of held-out data, across GPT-2 models and one Llama model. Reviewers agree the paper is well motivated and well written and that the methods introduced are practical and appropriate for the task.

Multiple reviewers have flagged, however, that the authors should evaluate the impact of this approach on downstream tasks, and we agree. We encourage the authors to evaluate on a suite of downstream evals - including HellaSwag, as the authors have mentioned, but even better would be a suite of downstream evals (e.g., Arc-Easy, Arc-Challenge, PiQA, BoolQ, possibly from an eval framework like mosaicml's llm-foundry or Eleuther AI's llm harness). This paper would be even stronger with evaluations that closed the loop to show when canonicalizing language models improves a practical downstream task.

Nevertheless, this is a strong submission and we recommend its acceptance for ICML.